# Variational Learning for Insertion-based Generation

**Yangtian Zhang** [* † 1]  **Zhe Wang** [* 2 3]  **Arthur Gretton** [2 3]  **Rex Ying** [1]  **David van Dijk** [1]
**Michalis K. Titsias** [2]  **Jiaxin Shi** [† 4]

## Abstract

Non-monotonic sequence generation methods, such as masked diffusion models, provide a flexible alternative to left-to-right autoregressive modeling by allowing tokens to be generated in non-fixed and prescribed orders. Despite their practical advantages, most existing non-monotonic models are order-agnostic and rely on a fixed-length grid, limiting their ability to support variable-length generation and adaptive insertion order. In this work, we introduce a probabilistic framework for learning insertion order in variable-length insertion models. We formalize a bijective correspondence between insertion trajectories and permutations, which enables an exact reparameterization of the data likelihood as a sum over permutations. Building on this result, we propose the **Insertion Process (IP)**, a stochastic generative model that jointly learns *where* to insert, *what* to insert, and *when* to terminate, trained via permutation-based variational inference. Unlike prior fixed-canvas approaches, IP natively supports variable-length generation and learns data-driven preferences over insertion orders. Experiments on goal-conditioned planning and molecular string generation demonstrate that learning insertion order improves both modeling quality and generalization in domains without a canonical left-to-right structure.

## 1. Introduction

Autoregressive sequence models typically generate tokens in a fixed left-to-right order (Bahdanau et al., 2014; Vaswani

---
[*]Equal contribution, randomized ordering. [†]Work done at Google DeepMind [1]Yale University [2]Google DeepMind [3]University College London [4]Meta Superintelligence Labs. Correspondence to: Yangtian Zhang <yangtian.zhang@yale.edu>, Zhe Wang <zhewang@google.com>, Michalis K. Titsias <mtit-sias@google.com>, Jiaxin Shi <ishijiaxin@gmail.com>.

*Proceedings of the 43ʳᵈ International Conference on Machine Learning*, Seoul, South Korea. PMLR 306, 2026. Copyright 2026 by the author(s).

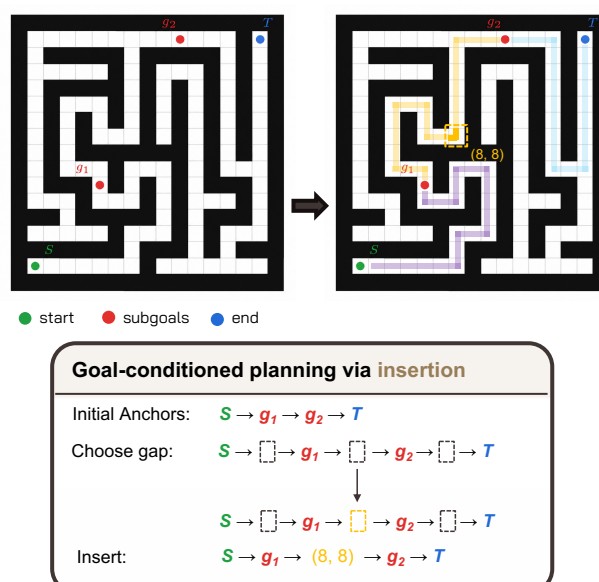

*Figure 1.* Goal-conditioned planning via insertion. Given initial anchors from start $S$ to subgoals $g_1, g_2$ and target $T$, the planner selects a gap and inserts a waypoint, producing a refined path.

et al., 2017). While effective, this factorization is not inherent to the data and can be misaligned with real-world generation problems (e.g., biological sequence design, program synthesis, and structured planning), where parts of the output may depend on future context. Enforcing a single left-to-right order in these settings can therefore be unnatural and inefficient.

A prominent alternative is *non-monotonic* sequence generation such as masked diffusion models (Shi et al., 2024; Sahoo et al., 2024; Ou et al., 2024), where a model constructs an output through randomly selecting and updating subsets of masked positions on a *fixed-length grid*. These methods enable parallel generation, infilling, and have shown strong performance across tasks. However, they still suffer from two limitations: **(i) Static canvas.** Masked discrete diffusion models typically operate on a prescribed, fixed-length set of masked positions, requiring an external length specification and making variable-length generation indirect. **(ii) Order agnosticism.** Their training objectives commonly marginalize over (or implicitly assume) many possible up-

date schedules, encouraging the network to be consistent with a combinatorial family of conditional distributions. In effect, the model is asked to approximate conditionals corresponding to a factorial number of generation orders, even though only a small subset of orders may align with the underlying structure of the data (Ni et al., 2026).

Recent work has begun to address these limitations, but largely in isolation. Insertion-based models remove the static canvas by generating variable-length sequences via token insertions at arbitrary positions (Stern et al., 2019; Gu et al., 2019a; Patel et al., 2025; Kim et al., 2025). Separately, learning-order methods introduce explicit distributions over permutations to learn data-dependent unmasking orders (Wang et al., 2025b). Yet existing learning-order approaches are typically developed in fixed-canvas masked/blank-filling settings, where global token positions serve as latent variables, and they do not directly extend to variable-length insertion. Combining these two directions is non-trivial: in insertion, *insertion locations* are not fixed coordinates but evolve with the partial sequence, so the generation order and the intermediate states are tightly coupled.

In this paper, we present a probabilistic framework for learning insertion order with varying sequence lengths, which allows for an unbiased estimate of the Evidence Lower Bound on the exact data likelihood. Our key observation is that, for a fixed target sequence, every valid insertion trajectory corresponds bijectively to a permutation of target indices; moreover, the intermediate states and relative insertion locations are deterministic functions of permutation prefixes. This change of variables converts marginalization over trajectory-dependent insertion actions into an exact sum over permutations, enabling variational training with a latent *order* variable. We instantiate this framework in the **Insertion Process (IP)** and show that learning data-dependent orders improves both modeling quality and generalization on planning benchmarks and molecular SMILES generation.

## 2. Variational Learning for Insertion Processes

In this section, we introduce **IP (Insertion Process)**, a probabilistic insertion model that jointly learns (i) a data-driven preference over insertion decisions (*where to insert*), (ii) a token distribution (*what to insert*), and (iii) a stopping rule (*when to terminate*) (Sec. 2.1). We train IP by maximizing an evidence lower bound (ELBO) on the data log-likelihood, treating the generation order as a latent variable and learning an amortized variational posterior over orders.

Direct variational inference over insertion order is nontrivial due to two issues: (i) *trajectory-dependent parameterization*: insertion actions are specified as *relative* slot indices

in the current partial sequence, making it difficult to define and approximate a stable posterior over order directly in the space of relative actions. (ii) *non-differentiable discrete optimization*: the latent order is discrete, making the variational objective non-differentiable.

To address these issues, we first exploit a bijection between relative insertion trajectories and global permutations of the target sequence (Sec. 2.2), which provides a fixed latent space and a permutation-marginalized likelihood for variational inference. We then optimize the resulting non-differentiable variational posterior using a Plackett–Luce family: permutations are sampled via the Gumbel–Top-$k$ trick and the variational parameters are trained with RE-INFORCE using a leave-one-out baseline (Sec. 2.3). On the generative side, we adopt a slot-conditional decoder that scores candidate insertion slots and predicts token content conditioned on the chosen slot, while handling variable length via an explicit termination decision (Sec. 2.4).

### 2.1. Process Formulation

To generate a sequence of variable length from a vocabulary $V$, we provide two variants of the Insertion Processes: these differ only in the ways they learn to terminate generation, which can be policy- or classifier-based. For simplicity of presentation, we use policy-based termination throughout the main text to present our core theoretical contributions, and provide details of the classifier-based termination variant in Section C.

---

**Insertion Process with Policy-based Termination.**
Let $y_0 = ()$ which is the initial state. At each generation step $i = 1, 2, \ldots$, the IP first samples the position of the insertion slot $z_i$ (1), and then decides the value $x_i$ to insert (2)

$$z_i \sim p_\phi(z \mid y_{i-1}), \quad z \in \{\text{TERM}, 1, \ldots, i\}, \quad (1)$$
$$x_i \sim p_\phi(x \mid y_{i-1}, z_i), \quad (2)$$

and define

$$y_i = \begin{cases} y_{i-1}, & z_i = \text{TERM}, \\ \text{insert}(y_{i-1}, z_i, x_i). \end{cases}$$

The process terminates when $z_i = \text{TERM}$ and outputs $y_{i-1}$.

---

Each $z_i$ is the *insertion* latent variable. It follows a policy distribution $p_\phi(z_i|y_{i-1})$ defined over $i + 1$ possible outcomes: the $i$ available insertion slots in $y_{i-1}$ and a special termination action TERM. This allows the model to dynamically decide whether to continue expanding the sequence or to stop generation based on the current state. If a valid

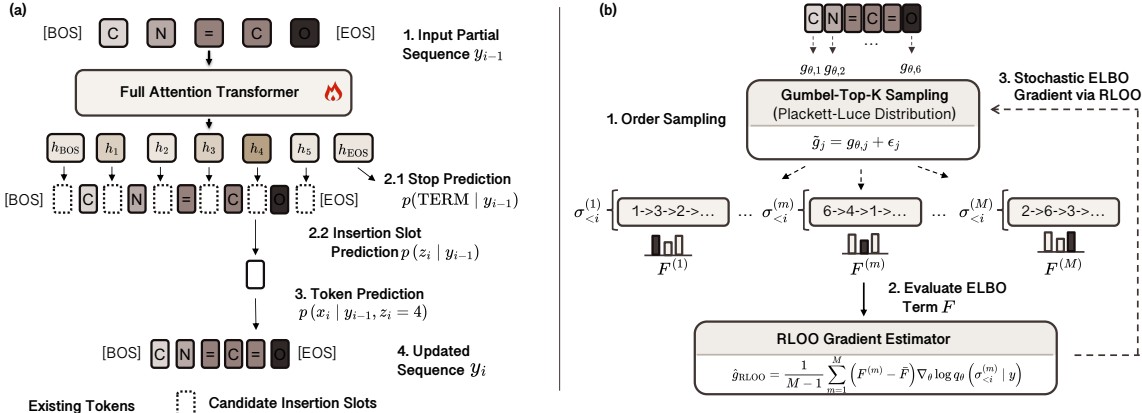

*Figure 2.* (**a**) **Generative Decoder Architecture.** At generation step $i$, the Transformer processes the partial sequence $y_{i-1}$ (augmented with boundary tokens) to produce contextual embeddings $h$. The embedding $h_{\text{EOS}}$ predicts the termination probability (Step 2.1). The remaining embeddings $h_k$ represent candidate insertion slots (dotted boxes), parameterizing the location distribution $p(z_i|y_{i-1})$ (Step 2.2). Conditioned on a selected slot (e.g., $z_i = 4$), the content head predicts the next token $x_i$ (Step 3). (**b**) **Policy-gradient optimization with Monte Carlo–estimated insertion orders** $q_\theta(\sigma_{<i} \mid y_L)$. The expectation over $q_\theta(\sigma_{<i} \mid y_L)$ in Eq. 12 is approximated using Gumbel–Top-$K$ sampling, and the resulting non-differentiable distribution is optimized via a RLOO gradient estimator.

slot is selected, $p_\phi(x_i|y_{i-1}, z_i)$ acts as a *classifier* over the vocabulary $V$ to determine the token content. The function $\text{insert}(y_{i-1}, z_i, x_i)$ denotes the operation that inserts $x_i$ into the chosen location, as illustrated in Figure 2.

For training, we would like to maximize the data log-likelihood $\log p_\phi(y_L)$. However, computing $\log p_\phi(y_L)$ requires marginalizing over the latent insertion trajectory (i.e., the sequence of slot decisions), which is generally intractable. We therefore optimize a variational lower bound (ELBO) by introducing a variational distribution over the latent variables. Since the relative slot indices $z_i$ are trajectory-dependent, we instead work with an equivalent *global* representation based on permutations of the target indices. The next subsection formalizes this reparameterization and shows how it leads to a convenient likelihood factorization for the ELBO.

### 2.2. Reparametrization for Likelihood Computation

Let $y_L = (y_{L,1}, \ldots, y_{L,L})$ be an observed sequence of length $L$. Under IP, producing $y_L$ corresponds to generating a trajectory of intermediate states and insertion actions that deterministically assemble $y_L$. A direct likelihood expression marginalizes over the trajectory variables, whose joint distribution factorizes as

$$p(\{y_i, x_i, z_i\}_{i=1}^L) = \prod_{i=1}^L p(y_i, x_i|y_{i-1}, z_i)p(z_i|y_{i-1}). \quad (3)$$

Each transition further decomposes as

$$p(y_i, x_i|y_{i-1}, z_i) = p(y_i|y_{i-1}, z_i, x_i) \, p(x_i|y_{i-1}, z_i), \quad (4)$$

where the state update is deterministic:

$$p(y_i|y_{i-1}, z_i, x_i) = \delta(y_i = \text{insert}(y_{i-1}, z_i, x_i)), \quad (5)$$

and $\delta(\cdot)$ denotes a Kronecker-delta point mass.

However, working directly in the trajectory space is inconvenient for variational inference because the insertion actions $z_i$ are *relative* to the current partial sequence: the meaning of "slot $k$" at step $i$ depends on which tokens have already been inserted and how they are arranged. Consequently, the sequence of relative actions does not live in a fixed coordinate system shared across trajectories that yield the same $y_L$. This makes it difficult to specify and learn a tractable variational approximation $q_\theta(z_{1:L}|y_L)$ with a stable parameterization.

We therefore change variables to a *global permutation* $\sigma \in S_L$, where $\sigma_i$ is the index of the element in $y_L$ that is filled at insertion step $i$. Intuitively, $\sigma$ records the order in which final positions are populated. Under this representation, the relative slot index becomes a deterministic function of the permutation prefix.

**Lemma 2.1** (Permutation–trajectory bijection). *Fix $y_L$. For any valid insertion trajectory that produces $y_L$, there exists a unique permutation $\sigma \in S_L$ such that the token inserted at step $i$ equals the final token at position $\sigma_i$, i.e. $x_i = y_{L,\sigma_i}$. Moreover, the relative insertion slot at step $i$ is determined by the prefix $\sigma_{\leq i}$ as*

$$z_i = f(\sigma_{\leq i}) := 1 + |\{ j \in \sigma_{<i} : j < \sigma_i \}|, \quad (6)$$

*i.e. $z_i$ is the rank of $\sigma_i$ among the sorted indices in $\sigma_{<i}$.* [1]

---

[1] In practice, $z_i$ can be computed via `argsort`. Let $s =$

For completeness and to fix notation in our setting, we provide a proof of Lemma 2.1 in Sec. A; related insertion encodings and more formal treatments appear in the combinatorics literature (e.g., Albert et al. 2005). A useful consequence is that $f(\sigma_{\leq i})$ depends only on the prefix $\sigma_{\leq i}$ (and not on $\sigma_{>i}$), which we later exploit to reduce variance in ELBO optimization.

Given $\sigma$, the trajectory is fully determined:

$$y_i = y_{L,\sigma_{\leq i}}, \qquad x_i = y_{L,\sigma_i}, \qquad z_i = f(\sigma_{\leq i}),$$

where $y_{L,\sigma_{\leq i}}$ denotes the subsequence of $y_L$ indexed by $\sigma_{\leq i}$ and then ordered by their original index.

To illustrate the mapping between insertion variables and permutations, consider the case where we always insert at the end of the data sequence, i.e., $z_i = i$. The corresponding permutation is the identity $\sigma = (1, 2, \ldots, L)$. In contrast, if each $z_i = 1$ (so we insert always in front of the sequence), the permutation becomes the reverse $\sigma = (L, L-1, \ldots, 1)$. Fig. 3 gives a further example which illustrates the evolution of the IP trajectory.

This reparameterization yields a permutation-marginalized likelihood.

**Theorem 2.2** (Permutation-marginalized likelihood)**.** *For any length-$L$ sequence $y_L$,*

$$p(y_L) = \sum_{\sigma \in S_L} p(y_L, \sigma). \qquad (7)$$

*Moreover, $p(y_L, \sigma)$ factorizes as*

$$\prod_{i=1}^{L} p_\phi(y_{L,\sigma_i} \mid y_{L,\sigma_{<i}}, f(\sigma_{\leq i})) \, p_\phi(f(\sigma_{\leq i}) \mid y_{L,\sigma_{<i}}), \quad (8)$$

*where $y_{L,\sigma_{<i}}$ denotes the subsequence of $y$ containing only the tokens at indices $\sigma_{<i}$, preserving their original order.*

The proof is provided in Appendix A.2. Eq. (7) makes explicit that the same observed $y_L$ can be generated by $L!$ distinct insertion orders, and replaces the trajectory-specific latent representation $(y, x, z)$ with a fixed latent space over permutations.

### 2.3. Training with Variational Inference

Given an observed $y_L$, a complete generative process also requires the model to stop after producing the $L$ tokens. We therefore train with the **augmented observation** $\{y_L, \text{AUX}\}$, specifically for policy-based termination $\text{AUX} := z_{L+1} = \text{TERM}$, which augments the policy with an extra dimension

---

sort$(\sigma_{\leq i})$ be the sorted prefix (ascending). Then $z_i$ is the 1-based position of $\sigma_i$ in $s$, i.e., $z_i = 1 + \text{where}(s = \sigma_i)$. Equivalently, if $r = \text{argsort}(\sigma_{\leq i})$ gives the indices that sort the prefix, then $z_i = 1 + \text{where}(r = i)$, since $\sigma_i$ is the last element of the prefix.

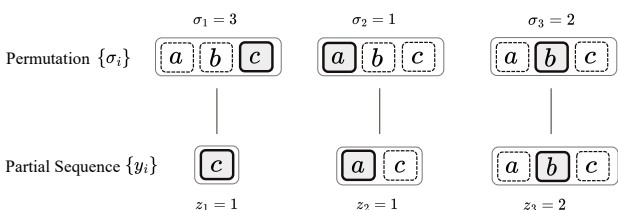

*Figure 3.* **Example of mapping from permutations to insertion orders in the IP model.** The final sequence is $y_3 = (a, b, c)$ and the permutation is $\sigma = (3, 1, 2)$. Then $\sigma_1 = 3 \Rightarrow z_1 = 1$, insert $x_1 = c$ into the empty sequence, $y_1 = (c)$; $\sigma_2 = 1 \Rightarrow z_2 = 1$, insert $x_2 = a$, $y_2 = (a, c)$; $\sigma_3 = 2 \Rightarrow z_3 = 2$, insert $x_3 = b$, $y_3 = (a, b, c)$.

to indicate termination, and for classifier-based termination $\text{AUX} := x_{L+1} = \text{EOS}$, which simply augments the vocabulary with one extra $\text{EOS}$ token. The full loss can then be formulated as:

$$\log p_\phi(y_L, \text{AUX}) = \log p_\phi(\text{AUX} \mid y_L) + \log p_\phi(y_L). \quad (9)$$

For both variants, the termination term $\log p_\phi(\text{AUX} \mid y_L)$ is tractable under our decoder and can be optimized directly. We detail the design of the two losses in Section B and Section C. The sequence likelihood $\log p_\phi(y_L)$ is intractable due to the sum over $L!$ permutations in Eq. (7), and we optimize a variational lower bound.

**Permutation ELBO.** By Lemma 2.1, every valid insertion trajectory that constructs $y_L$ corresponds bijectively to a *permutation* $\sigma \in S_L$ specifying the order in which final positions are filled. We therefore treat $\sigma$ as the latent variable representing the (otherwise trajectory-dependent) insertion order, and introduce an amortized variational posterior $q_\theta(\sigma \mid y_L)$ to maximize the variational lower bound of $\log p_\phi(y_L)$ (Theorem 2.2):

$$\log p_\phi(y_L) \geq \mathcal{L}_{\text{ELBO}}(y_L) := \sum_{\sigma \in S_L} q_\theta(\sigma \mid y_L) \log \frac{p_\phi(y_L, \sigma)}{q_\theta(\sigma \mid y_L)}. \quad (10)$$

We parameterize $q_\theta(\sigma \mid y_L)$ as a Plackett–Luce (PL) model,

$$q_\theta(\sigma \mid y_L) = \prod_{i=1}^{L} q_\theta(\sigma_i \mid \sigma_{<i}, y_L), \quad (11)$$

which provides a tractable, prefix-conditioned family for orders and enables efficient sampling (e.g., via Gumbel–Top-$k$). Using Theorem 2.2 and (11), the ELBO can be written as

$$\sum_{i=1}^{L} \mathbb{E}_{q_\theta(\sigma_{<i}|y_L)} \Big[ \mathbb{E}_{q_\theta(\sigma_i|\sigma_{<i}, y_L)} [\log p_\phi(y_{L,\sigma_i} | f(\sigma_{\leq i}), y_{L,\sigma_{<i}})$$
$$+ \log p_\phi(f(\sigma_{\leq i}) | y_{L,\sigma_{<i}}) - \log q_\theta(\sigma_i|\sigma_{<i}, y_L)] \Big]. \quad (12)$$

**Algorithm 1** Training IP with permutation VI and RLOO

---

1: **Given:** dataset $\mathcal{D}$, generative decoder $p_\phi$, inference model $q_\theta$, #samples $M$
2: **while** training **do**
3:     Sample $y_L \sim \mathcal{D}$
4:     Sample $M$ permutations $\{\sigma^m\}_{m=1}^M \sim q_\theta(\cdot \mid y_L)$ through the Gumbel–Top-$k$ trick.
5:     Sample $i \sim \text{Unif}\{1, \ldots, L+1\}$
6:     **for** $m = 1, \ldots, M$ **do**
7:         Form the prefix $\sigma_{<i}^m$ and partial sequence $y_{L,\sigma_{<i}^m}$
8:         Compute $F^m$ using the exact $\mathbb{E}_{q(\sigma_i|\sigma_{<i}^m, y_L)}[\cdot]$ (14)
9:     **end for**
10:    Form the $M$-sample RLOO objective (19) or (24) and update $(\phi, \theta)$
11: **end while**

---

**Variance-reduced ELBO estimation and optimization.** Eq. (12) involves expectations over a discrete latent order, so naïvely sampling full permutations yields a high-variance estimator, and the resulting samples are non-differentiable w.r.t. the variational parameters $\theta$. We exploit the permutation–trajectory structure to reduce variance: since the induced slot index $f(\sigma_{\leq i})$ depends only on the prefix $\sigma_{\leq i}$ (Lemma 2.1), the inner expectation over the next index $\sigma_i$ conditioned on a fixed prefix $\sigma_{<i}$ can be evaluated exactly. Concretely, for fixed $\sigma_{<i}$ we define

$$F(\sigma_{<i}; i, y_L) := \mathbb{E}_{q_\theta(\sigma_i|\sigma_{<i}, y_L)}[\,\cdot\,], \qquad (13)$$

where $[\cdot]$ denotes the bracketed log-ratio term in Eq. (12), and compute it in closed form as

$$F(\sigma_{<i}; i, y_L) = \sum_{j \notin \sigma_{<i}} q_\theta(j \mid \sigma_{<i}, y_L) \, [\,\cdot\,]_{\sigma_i = j}. \qquad (14)$$

The remaining outer expectation over prefixes $\mathbb{E}_{q_\theta(\sigma_{<i}|y_L)}$ is still intractable and non-differentiable, so we estimate it by Monte Carlo using samples from the PL posterior (implemented via Gumbel–Top-$k$), and optimize $\theta$ with **REINFORCE Leave-One-Out** (RLOO, Kool et al., 2019a). For efficiency, we subsample a single time step $i \sim \text{Unif}\{1, \ldots, L+1\}$ to form an unbiased stochastic estimator of the sum over $i$ (Algorithm 1; Appendix B).

## 2.4. Network Architectures

We parameterize the Insertion Process using an encoder-decoder architecture. The generative model (decoder) $p_\phi$ predicts the next insertion slot $z$ and the corresponding token $x$, while the inference network (encoder) $q_\theta$ approximates the posterior over the permutation $\sigma$.

**Generative Decoder.** To generate a sequence, the decoder must predict both *where* to insert a token and *what* token to insert.

For policy-based termination IP, during training, given a partial sequence at step $i$: $y_{i-1} = (t_1, \ldots, t_{i-1})$, we construct the augmented input sequence $y_{i-1}^{\text{aug}} = (\text{BOS}, t_1, \ldots, t_{i-1}, \text{EOS})$. A Transformer decoder processes this input to produce contextualized embeddings $\mathbf{h}_0, \mathbf{h}_1 \ldots, \mathbf{h}_i$. We define the representation of the $k$-th insertion slot as the embedding of the token immediately preceding it, $\mathbf{h}_{k-1}$. Specifically, $\mathbf{h}_0$ (the embedding of BOS) represents the first slot, and $\mathbf{h}_j$ represents the slot immediately following token $t_j$. We use $\mathbf{h}_{\text{EOS}} \triangleq \mathbf{h}_i$ to represent the termination of the generation process, as illustrated in Fig. 2. For classifier-based termination, the EOS token is not appended to the input sequence, as termination is determined by the classifier's prediction of EOS.

The joint distribution at step $i$ is factorized as $p_\phi(x_i, z_i|y_{i-1}) = p_\phi(x_i|z_i, y_{i-1})p_\phi(z_i|y_{i-1})$. First, a *location head* computes the probability of selecting slot $k$[2] via a softmax over the embeddings $\mathbf{h}_{0:i}$:

$$p_\phi(z_i = k|y_{i-1}) = \frac{\exp(\mathbf{w}_z^\top \mathbf{h}_{k-1})}{\sum_{j=0}^i \exp(\mathbf{w}_z^\top \mathbf{h}_j)}. \qquad (15)$$

Conditioned on the selected slot $z_i = k$, a *content head* predicts the token distribution using that slot's embedding:

$$p_\phi(x_i = \cdot|z_i = k, y_{i-1}) = \text{Softmax}(\mathbf{W}_x \mathbf{h}_{k-1}). \qquad (16)$$

Note that, to allow for flexible length generation during sampling, the generative decoder is conditioned only on the partial sequence $y_i$, not on $y_L$ nor $L$.

**Inference Encoder (Plackett-Luce Model).** The encoder $q_\theta$ parameterizes a distribution over the global permutation $\sigma$ of the observed sequence $y_L$. We model $q_\theta(\sigma|y_L)$ as a Plackett-Luce model, where the probability of the permutation is calculated as follows.

A bidirectional Transformer encoder processes the full sequence $y_L$ to output a score $g_{\theta,j}$ for each global index $j \in \{1, \ldots, L\}$. Ideally, we would sample $\sigma$ sequentially such that $q(\sigma_i|\sigma_{<i}, y_L) \propto \exp(g_{\theta,\sigma_i})$. To enable efficient sampling without sequential recurrence during training, we utilize the **Gumbel-Top-$k$ trick** (Kool et al., 2019b). We sample i.i.d. Gumbel noise $\epsilon_j \sim \text{Gumbel}(0, 1)$ for each token and compute perturbed scores:

$$\tilde{g}_j = g_{\theta,j} + \epsilon_j. \qquad (17)$$

The sampled permutation $\sigma$ is obtained by sorting the indices in descending order of $\tilde{g}_j$. This effectively draws a sample from the Plackett-Luce distribution defined by scores $g_\theta$. Finally, $q_\theta(\sigma \mid y_L)$ is obtained through taking `softmax` over the perturbed scores. Crucially, under this

---

[2]Selecting $k = i + 1$ means termination.

parameterization, the relative insertion position $z_i$ is not sampled directly by the encoder but is derived deterministically from the sampled global permutation $\sigma$ via the mapping $z_i = f(\sigma_{\leq i})$.

We provide additional detail of our training and experiment setup in Section D.

## 3. Related Work

**Non-monotonic generation.** Non-monotonic generation is widely studied in discrete diffusion and any-order autoregressive modeling (Uria et al., 2014). Discrete diffusion models such as masked diffusions (Austin et al., 2021; Shi et al., 2024; Sahoo et al., 2024; Campbell et al., 2022; Gat et al., 2024; Lou et al., 2023) define a forward masking process over discrete states and train a reverse denoiser to unmask them. These models commonly operate on a fixed canvas and can impose conditional-independence approximations that affect coherence. LO-ARM (Wang et al., 2025b) goes further by learning a state-dependent unmasking policy with a variational objective, demonstrating strong results on domains such as graphs where canonical ordering is ambiguous; however, it is also formulated over fixed-size sequences, and does not directly model variable-length generation through an explicit insertion-and-termination mechanism.

**Insertion-based generative models.** Insertion-based generation constructs outputs by repeatedly selecting an insertion slot and inserting a token, rather than appending tokens strictly left-to-right. Early insertion decoders show that permitting arbitrary insertion orders improves flexibility: Gu et al. (2019a) enables non-monotonic decoding and can recover a generation order at inference time by adaptively searching over insertion trajectories (e.g., beam search). Other work treats the generation order as latent. Li et al. (2021) uses variational inference to infer non-monotonic orderings as global permutations, but relies on permutation-structured relaxations from combinatorial optimization (e.g., Sinkhorn-style constructions) and Bethe permanent approximation, which can introduce nontrivial computational overhead and approximation bias.

A complementary thread develops *order-agnostic* insertion models, emphasizing flexible schedules and efficient decoding rather than explicitly learning an instance-specific order posterior. The Insertion Transformer (Stern et al., 2019) supports arbitrary insertion orders (including balanced-tree schedules) and can reduce decoding iterations. KERMIT (Chan et al., 2019) frames insertion as a unified mechanism for modeling joint distributions and multiple conditional factorizations within a single network. Other work expands the edit space beyond insertion: the Levenshtein Transformer (Gu et al., 2019b) and Edit-flow (Havasi et al.,

2025) incorporate both insertion and deletion to enable iterative refinement. More recently, Insertion Language Models (Patel et al., 2025) revisit single-token arbitrary-position insertions, highlighting advantages in constraint-heavy generation and variable-length infilling. Our work complements these by introducing a likelihood-based insertion process with explicit termination and an ELBO that learns context-dependent generation order from data.

A related family couples non-monotonic expansion with blanks or masks. Blank Language Models (Shen et al., 2020) generate by introducing blanks and filling them, effectively choosing where to expand next. DreamOn (Wu et al., 2025) and FlexMDM (Kim et al., 2025) enable any-order generation within a masked-diffusion framework by allowing mask insertion followed by unmasking. Compared to blank/mask expansion where generation proceeds through intermediate placeholder states, our formulation directly models token insertions as the primitive stochastic actions in a single insertion trajectory. Moreover, FlexMDM shares the limitation of order-agnostic methods, lacking a policy to prioritize fitting instance-specific generation orders.

## 4. Experiments and Analysis

Our experiments are designed to answer the following questions:

- Whether insertion-based generation outperforms standard autoregressive models on generating sequences that are not natural left-to-right.

- Whether learning an instance-specific insertion policy improves upon order-agnostic methods.

- Whether these advantages translate to better goal-conditioned planning and improved performance in flexible-length conditional generation tasks.

To study these questions, we consider two representative domains. We first evaluate on **planning tasks**, where solutions must satisfy global constraints or coordinate multiple subgoals whose natural construction order may vary across instances. We then evaluate on **molecular string generation**, where sequences are induced by graph traversal procedures and admit multiple valid linearizations, making left-to-right generation orders largely arbitrary.

To evaluate the IP on the above two benchmarks, we introduce the following baselines: 1) Fixed-Order Autoregressive Models (FO-ARMs), 2) Any-Order ARMs (AO-ARMs) (Uria et al., 2014), 3) Masked Diffusion Models (MDMs) and 4) Learning-Order ARMs (LO-ARMs) (Wang et al., 2025b), based on the following rationale. First, FO-ARMs can be viewed as insertion processes with fixed insertion ordering, which always insert at the end of a partial

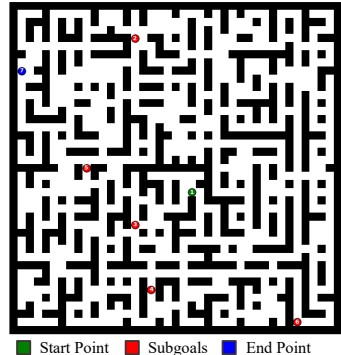

■ Start Point  ■ Subgoals  ■ End Point

*Figure 4.* A perfect maze.

*Table 1.* Performance on Maze Planning benchmarks (mean). The best result is **bolded** and the second-best is underlined.

| Method | BRAIDED | | | IMPERFECT | | | PERFECT | | |
|---|---|---|---|---|---|---|---|---|---|
| | EASY | MED | HARD | EASY | MED | HARD | EASY | MED | HARD |
| *Monotonic (left-to-right) baselines* | | | | | | | | | |
| FO-ARM | 96.2 | 94.2 | 93.0 | 92.5 | 88.2 | 76.9 | 96.2 | 58.3 | 27.2 |
| *Non-monotonic baselines* | | | | | | | | | |
| MDM | 51.9 | 59.0 | 60.7 | 56.1 | 51.7 | 50.0 | 12.2 | 0.2 | 0.0 |
| AO-ARM | 54.2 | 61.8 | 65.4 | 60.3 | 55.2 | 51.3 | 11.7 | 1.1 | 0.0 |
| LO-ARM | 67.7 | 61.3 | 56.4 | 79.8 | 60.1 | 51.6 | 13.8 | 32.0 | 25.2 |
| FlexMDM | 86.9 | 86.7 | 89.5 | 78.5 | 83.1 | 84.2 | 0.1 | 0.0 | 0.0 |
| **IP** (Ours) | **100.0** | **100.0** | **99.9** | **100.0** | **98.4** | **95.3** | **99.4** | **98.1** | **97.9** |

*Table 2.* Performance on Star Graph Planning Benchmark. Accuracy is reported in %. Lower is better for distance metrics.

| Model | Seq. Acc. | Tok. Acc. | Ham. Dist.↓ | Lev. Dist.↓ |
|---|---|---|---|---|
| *Monotonic (left-to-right) baselines* | | | | |
| FO-ARM | 24.0 | 40.6 | 9.58 | 9.44 |
| *Non-monotonic baselines* | | | | |
| MDM | 25.0 | 66.1 | 6.03 | 5.53 |
| AO-ARM | 26.5 | 69.3 | 5.46 | 5.12 |
| LO-ARM | 30.1 | 72.5 | 4.91 | 4.16 |
| FlexMDM | 0.0 | 14.6 | 14.30 | 14.29 |
| **IP** (Ours) | **83.0** | **85.7** | **3.57** | **2.44** |

sider PERFECT, IMPERFECT, and BRAIDED maze families with increasing structural flexibility, and easy/medium/hard tiers by increasing grid size, obstacle density, and the number of targets/subgoals.

The second benchmark is **star-graph planning** (Patel et al., 2025). The input prefix consists of a randomly ordered list of directed edges (node pairs), followed by the source node, the goal node, and a special graph-BOS token; the target trajectory is the directed edge sequence forming the path from source to goal (details in Appendix D.2). We report sequence accuracy and token accuracy, along with edit-based distances (Hamming and Levenshtein) to quantify partial deviations from the gold path.

As shown in Table 1, IP achieves the strongest performance across all three maze families and remains robust as difficulty increases. The advantage is most evident on PERFECT mazes (especially HARD), where the solution space is most constrained and fixed-order baselines degrade. On star graphs (Table 2), IP substantially improves both sequence-level and token-level accuracy and reduces edit distances to the target path. Overall, these results suggest that learning *where to insert* during generation is particularly beneficial when planning decisions depend on long-range constraints or instance-specific construction orders.

sequence, i.e., generating from left-to-right. Through comparing the IP against FO-ARMs, we want to see whether IP can learn meaningful insertion orderings while maintaining competitive performance. Second, AO-ARMs, LO-ARMs, and MDMs generate new samples in given canvases of fixed sizes or lengths. Through comparing these two types of generative models, we want to show that IP can also yield better generation performance via flexible-length generation without prescribing canvas sizes.

## 4.1. Planning Tasks

We evaluate insertion-based generation on two synthetic planning benchmarks where solutions must satisfy global constraints and there is no single natural left-to-right construction order. In both benchmarks, each instance is represented as a single 1D token sequence, and the learning problem is to generate a variable-length *trajectory* segment conditioned on a task-specific prefix.

The first benchmark is **maze planning** (Kim et al., 2025). Each example is a grid-maze path flattened into a token sequence by mapping each visited cell to a discrete token (a flattened cell index), making it a standard sequence modeling problem. Conditional generation is defined by providing an ordered subset of visited states as subgoals (full construction and conditioning in Appendix D.1); the model must generate a path that visits these subgoals in order. We con-

## 4.2. Molecular String Generation

SMILES strings (Weininger, 1988) are linearizations of molecular graphs: while a left-to-right SMILES can often be interpreted as a depth-first traversal, the traversal choices (and the placement of branch/ring annotations) are not unique, and correct strings must satisfy global pairing constraints—parentheses for branches and digit tokens for ring closures. These long-range dependencies make SMILES a natural testbed for non-monotonic generation, where a model can decide global structure early and then refine local content.

We evaluate unconditional SMILES generation on **GuacaMol** (Brown et al., 2019), reporting both per-sample qual-

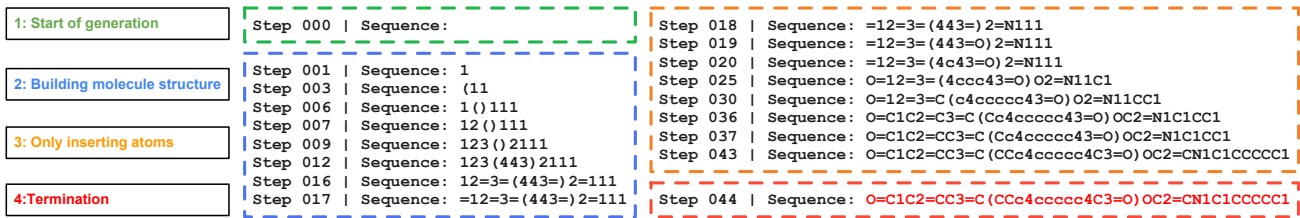

*Figure 5.* An example of generating a SMILES sample via the Insertion Process. The learned generation process is phased into four stages: 1) At Stage 1 (Step 0), the model starts with an empty string. 2) From Step 1 to 17 in Stage 2, the model first lays out the molecule's skeleton by generating only matching parentheses and digit pairs (ring-closure markers). 3) In Stage 3, the model then inserts atom symbols into this scaffold to form the full SMILES string. 4) Finally, at Step 44, the model signals termination, and the generated molecule stays the same as that in Step 43.

*Table 3.* Unconditional molecule generation performance on the GuacaMol SMILES benchmark. We evaluated on the following metrics: **V**alidity, **U**niqueness, **N**ovelty, FCD and KL divergence. **V.U.** means both valid and unique, and **V.U.N.** means samples are valid, unique and novel. The metrics are calculated on samples generated by each method. Bold and underlined numbers indicate the best and second-best results, respectively. [†]Results taken from Brown et al. (2019).

| Model | V. ↑ | V.U. ↑ | V.U.N. ↑ | KL div ↑ | FCD ↑ |
|---|---|---|---|---|---|
| Training set | 100.0 | 100.0 | 0.0 | 99.9 | 92.8 |
| LSTM[†] | 95.9 | 95.9 | 87.5 | 99.1 | **91.3** |
| VAE[†] | 87.0 | 86.9 | 84.7 | 98.2 | 86.3 |
| AAE[†] | 82.2 | 82.2 | 82.1 | 88.6 | 52.9 |
| FO-ARM (Transformer) | **98.5** | **98.3** | 80.4 | **99.4** | 90.3 |
| AO-ARM | 83.1 | 83.1 | 80.7 | 93.1 | 69.3 |
| LO-ARM | 89.7 | 89.4 | 81.6 | 96.1 | 83.7 |
| AO-IP | 88.2 | 88.2 | 87.7 | 95.3 | 79.1 |
| IP (policy-based term.) | 97.2 | 97.0 | 94.9 | 97.1 | 88.2 |
| IP (classifier-based term.) | 97.4 | 97.3 | **95.6** | 97.2 | 89.2 |

*Table 4.* Evaluation on the four conditional generation tasks. For each task, we generate 1024 samples and evaluate the samples on **V**alidity and **U**niqueness.

| Task | V. ↑ | | V.U. ↑ | |
|---|---|---|---|---|
| | IP | AO-IP | IP | AO-IP |
| Frag. decoration | **99.8** | 40.9 | **10.4** | 5.1 |
| Linker design | **99.5** | 50.6 | **13.1** | 8.2 |
| Linker + partial decoration | **99.9** | 42.8 | **24.0** | 20.6 |
| Linker + full decoration | **99.9** | 41.3 | **34.2** | 31.1 |

ity (Validity, Uniqueness, Novelty) and distributional similarity to the test set (KL divergence and Fréchet ChemNet Distance; definitions in Sec. D.4). All sequence baselines use comparable Transformer backbones. We reimplement the FO-ARM and tune it to match the distributional performance reported for Transformer FO-ARM models on GuacaMol (Wang et al., 2025a). For IP, we additionally compare against a variant with a fixed random insertion policy (Any-Order IP, or AO-IP) to assess the importance of learning the insertion policy.

**Interpretable SMILES grammar from learned insertion schedules.** Despite having no hand-crafted SMILES rules as prior knowledge, IP learns a consistent, interpretable generation order (Fig. 5). Specifically, starting with an empty string, the typical learned process is: 1) Build the molecular structure (rings and connections) by first generating digit tokens for ring enclosures and cuts and proposing substructures via pairs of matching parentheses. 2) Insert atoms. 3) Signal termination. Across 100,000 unconditional samples, 99.7% follow this pattern, suggesting that the model has internalized a stable notion of "SMILES well-formedness".

**Unconditional generation results and a flexibility–fidelity trade-off.** FO-ARM/LSTM appends tokens strictly at the end of a sequence, whereas IP permits insertions at dynamically expanding slots whose number grows linearly with the generation steps. Table 3 highlights a clear trade-off between *distribution matching* and *chemical space exploration*. FO-ARM and LSTM achieve the best distributional scores (KL and FCD). IP, however, trails FO-ARM/LSTM slightly on distributional metrics yet achieves the highest novelty ($\sim 20\%$ higher than FO-ARM). Furthermore, comparing IP to its random-order ablation (AO-IP) underscores the necessity of learning the insertion order. While AO-IP maintains high novelty, its validity drops to 88.2%. This indicates that random insertions frequently violate chemical syntax, whereas IP successfully internalizes the structural rules needed to produce valid molecules. Finally, we observe that the classifier-based termination variant of IP consistently outperforms the policy-based variant in this task.

Crucially, the combination of high validity and novelty suggests that IP is not merely memorizing training molecules; rather, it explores the same underlying chemical distribution while generating structurally diverse samples. Such behavior is particularly desirable for molecule discovery, where expanding the set of plausible candidates is often more valuable than tightly matching the empirical distribution.

**Slot-constrained conditional generation: decoration and linker design.** Finally, we demonstrate the IP's flexible-

length generation via molecule decoration and linker design. Specifically, we initialize the process with a partial SMILES fragment and restrict which insertion slots are allowed, while leaving the number of insertions unconstrained—generation terminates whenever the model predicts `TERM`. This is difficult to realize with a left-to-right FO-ARM, especially for constraints that require inserting *inside* (or *between*) conditioned fragments (e.g., linker design).

We consider four constraint patterns (Table 4): (1) **Fragment completion + decoration:** condition on an incomplete benzene fragment `1ccccc1` and allow insertions at both ends. (2) **Linker design:** condition on two separated fragments `c1ccccc12ccccc2` and allow insertions only in the middle slot between them. (3) **Linker + partial decoration:** allow insertions in the middle of the two fragments and at one end. (4) **Linker + full decoration:** allow insertions in the middle and at both ends. Note that in each task, one fragment is incomplete, e.g., `2ccccc2` in Task (2), rendering the concatenated SMILES invalid.

Using the GuacaMol model pre-trained for unconditional generation, we sample 1024 candidates per task and report Validity/Uniqueness. Across all tasks, validity remains near 100%, indicating the model can reliably complete constrained fragments and terminate upon completion. When we allow a larger conditional solution space through relaxing the constraints on insertion slots, uniqueness increases substantially. The comparison with AO-IP shows that learning the insertion policy also improves conditional generation performance. We include qualitative samples in Sec. F (Tasks 1–4).

## 5. Conclusion

We have presented an unbiased variational inference method for training a transformer-based insertion generative model. Despite the non-parametric nature of the insertion process, we derived an analytic reparametrization of the log-likelihood that permits the use of effective permutation-based variational inference. Results on two different domains, planning and molecule string generation, demonstrated that our method can be effective in practice. One direction for future work is to apply our method to other generative modeling domains, with no canonical left-to-right orderings, such as protein design.

## Impact Statement

This paper presents work whose goal is to advance the field of Machine Learning. There are many potential societal consequences of our work, none of which we feel must be specifically highlighted here.

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

# A. Proofs

### A.1. Lemma 2.1

The equivalence between permutation and insertion order is a well-studied problem (Albert et al., 2005). Here, we provide a proof of Lemma 2.1 to make the work self-contained.

Let us denote $z = (z_1, \ldots, z_L)$ the insertion sequence and $x = (x_1, \ldots, x_L)$ the corresponding token sequence. We first construct the inverse function $\sigma = f^{-1}(z)$ that gives a unique permutation $\sigma = (\sigma_1, \ldots, \sigma_L)$ given $z$, such that $x_i = y_{L,\sigma_i}$. Then we show that this function is invertible so that $z = f(\sigma)$.

Based on the IP iteration, each $x_i$ is inserted at the $i$-step in location $z_i$ where $z_i$ specifies the exact global location of $x_i$ in the current string $y_i$. Then as the string grows this global location of $x_i$ changes according to the recursion

$$\text{Initialization: } \sigma_i^{(i)} = z_i$$
$$\text{Iteration: } \sigma_i^{(j)} = \sigma_i^{(j-1)} + I(z_j \le \sigma_i^{(j-1)}), \; j = i+1, \ldots, L$$

where the indicator function $I(z_j \le \sigma_i^{(j-1)})$ increments the global locations of $x_i$ whenever each new token $x_j$ (with $j > i$) is inserted on the left of $x_i$. At the final iteration $\sigma_i := \sigma_i^{(L)}$ is precisely the global location of $x_i$ in $y_L$, i.e., $x_i = y_{L,\sigma_i}$. In summary we can denote the above recursive mappings as $\sigma_i = f^{-1}(z_{\ge i}), i = 1, \ldots, L$, or in a vectorized form as $\sigma = f^{-1}(z)$. Then, for this constructed $\sigma$ we can easily see that in the opposite direction the mappings are

$$z_i = \text{index}(\text{sort}(\sigma_{\le i}) = \sigma_i) = f(\sigma_{\le i}), \; i = 1, \ldots, L,$$

or $z = f(\sigma)$ in short. Thus, we have constructed the mappings in both directions which completes the proof.

### A.2. Theorem 2.2

We start from $p(y_L)$ as computed by IP:

$$p(y_L) = \sum_{\{y_i, x_i, z_i\}_{i=1}^{L-1}, (x_L, z_L)} p(\{y_i, x_i, z_i\}_{i=1}^{L}) = \sum_{z} \sum_{\{y_i, x_i,\}_{i=1}^{L-1}, x_L} p(\{y_i, x_i, z_i\}_{i=1}^{L})$$

Using Lemma 2.1 we change variables based on $z_i = f(\sigma_{\le i})$ and write the above as

$$p(y_L) = \sum_{\sigma} \sum_{\{y_i, x_i,\}_{i=1}^{L-1}, x_L} p(\{y_i, x_{f(\sigma_{\le i})}, f(\sigma_{\le i})\}_{i=1}^{L}).$$

which requires summing over the permutations $\sigma$. Now due to the delta masses $\delta(y_i = \text{insert}(y_{i-1}, f(\sigma_{\le i}), x_i))$ (each equals one if $x_i$ is correctly inserted in $y_{i-1}$ at slot $f(\sigma_{\le i})$ and zero otherwise) that appear in the joint $p(\{y_i, x_i, f(\sigma_{\le i})\}_{i=1}^{L})$ the inner sum $\sum_{\{y_i, x_i,\}_{i=1}^{L-1}, x_L}$ simplifies as

$$\sum_{\{y_i, x_i,\}_{i=1}^{L-1}, x_L} p(\{y_i, x_i, f(\sigma_{\le i})\}_{i=1}^{L}) = \prod_{i=1}^{L} p(y_{L,\sigma_i} | y_{L,\sigma_{<i}}, f(\sigma_{\le i})) p(f(\sigma_{\le i}) | y_{L,\sigma_{<i}}) = p(y_L, \sigma)$$

since only a single term in the sum, i.e., the term with the insertion path $\{y_i = y_{L,\sigma_{\le i}}, x_i = y_{L,\sigma_i}, z_i = f(\sigma_{\le i})\}_{i=1}^{L}$, can be non-zero. Therefore, we conclude that

$$p(y_L) = \sum_{\sigma} p(y_L, \sigma),$$

which completes the proof.

# B. Optimization of the Training Objective with Policy-based Termination

For model training the Maximum Likelihood approximate objective per data sequence $y_L$ is

$$\log p(z_{L+1} = \text{TERM} | y_L) + \sum_{i=1}^{L} \mathbb{E}_{q(\sigma_{<i}|y_L)} \left[ \mathbb{E}_{q(\sigma_i|\sigma_{<i}, y_L)} \left[ \log \frac{p(y_{L,\sigma_i} | f(\sigma_{\le i}), y_{L,\sigma_{<i}}) \, p(f(\sigma_{\le i}) | y_{L,\sigma_{<i}})}{q(\sigma_i | \sigma_{<i}, y_L)} \right] \right] \tag{18}$$

or

$$\sum_{i=1}^{L+1} \left\{ 1(i = L+1) \log p(\text{TERM}|y_L) + 1(i \le L) \mathbb{E}_{q(\sigma_{<i}|y_L)} \left[ \mathbb{E}_{q(\sigma_i|\sigma_{<i},y_L)} \log \frac{p\left(y_{L,\sigma_i}|f(\sigma_{\le i}), y_{L,\sigma_{<i}}\right) p\left(f(\sigma_{\le i})|y_{L,\sigma_{<i}}\right)}{q(\sigma_i|\sigma_{<i},y_L)} \right] \right\}$$

Then, in practice for computationally efficient optimization we randomly sample one term $i$ in the sum, and also $\sigma_{<i} \sim q(\sigma_{<i}|y_L)$. This gives the stochastic unbiased objective

$$(L+1) \left\{ 1(i = L+1) \log p(\text{TERM}|y_L) + 1(i \le L) \mathbb{E}_{q(\sigma_i|\sigma_{<i},y_L)} \left[ \log \frac{p\left(y_{L,\sigma_i}|f(\sigma_{\le i}), y_{L,\sigma_{<i}}\right) p\left(f(\sigma_{\le i})|y_{L,\sigma_{<i}}\right)}{q(\sigma_i|\sigma_{<i},y_L)} \right] \right\}$$

To perform unbiased gradient-based maximization and deal with the expectation over $q(\sigma_{<i}|y_L)$ we apply REINFORCE Leave-One-Out (Kool et al., 2019a). Specifically, we use two independent permutation samples $\sigma^1, \sigma^2 \sim q(\cdot|y_L)$ (i.e. $M = 2$)and implement the objective

$$\frac{L+1}{2} \left\{ \left( \log q(\sigma_{<i}^1|y_L) - \log q(\sigma_{<i}^2|y_L) \right) \text{stopgrad}[F^1 - F^2] + F^1 + F^2 \right\}, \tag{19}$$

where for $j = 1, 2$

$$F^j = 1(i = L+1) \log p(\text{TERM}|y_L) + 1(i \le L) \mathbb{E}_{q(\sigma_i^j|\sigma_{<i}^j,y_L)} \left[ \log \frac{p\left(y_{L,\sigma_i^j}|f(\sigma_{\le i}^j), y_{L,\sigma_{<i}^j}\right) p\left(f(\sigma_{\le i}^j)|y_{L,\sigma_{<i}^j}\right)}{q(\sigma_i^j|\sigma_{<i}^j,y_L)} \right]$$

With the help of stopgrad$[F^1 - F^2]$ this allows automatic differentiation to return the unbiased gradient. To monitor convergence we compute the two-sample stochastic objective

$$\mathcal{L} = \frac{L+1}{2} \left( F^1 + F^2 \right). \tag{20}$$

For a minibatch these stochastic objectives are further averaged over the minibatch, where each data sequence $y_{L_n}$ has its own length $L_n$.

## C. Insertion Process with Classifier-based Termination

We define the Insertion Process with classifier-based termination as follows:

> **Insertion Process with Classifier-based Termination**   Let $y_0 = ()$ which is the initial state. At each generation step $i = 1, 2, \ldots$, the IP first samples the insertion slot and then samples its value
>
> $$z_i \sim p(z \mid y_{i-1}), \qquad i \in \{1, \ldots\},$$
> $$x_i \sim p(x \mid y_{i-1}, z_i),$$
>
> and define
>
> $$y_i = \text{insert}(y_{i-1}, z_i, x_i).$$
>
> The process terminates when $x_i = \text{EOS}$ and outputs $y_i$.

Specifically, during training, each $y_L$ is augmented with an extra End-of-Sequence (EOS) token to $\{y_L, x_{L+1} = \text{EOS}\}$, and therefore the effective sequence length becomes $L + 1$. Note that, with the classifier-based termination, we use the same network $p(x \mid y_{i-1}, z_i)$ to sample $x_i$ even when generating EOS: there is no need for a separate network to predict termination, as in the case of policy-based IP. This simplification yields a better model and sample efficiency as we have seen in Table 3, under the training protocol described below (Section C.1).

Similar to Equation (18), the training loss for the classifier-based (CB) IP is written as

$$\mathcal{L}^{\text{CB}} = \mathcal{L}_{\text{Termination}}^{\text{CB}} + \mathcal{L}_{\text{ELBO}}^{\text{CB}}$$

$$= \log p(x_{L+1} = \text{EOS}|y_L) \quad + \sum_{i=1}^{L} \mathbb{E}_{q(\sigma_{<i}|y_L)} \left[ \mathbb{E}_{q(\sigma_i|\sigma_{<i},y_L)} \left[ \log \frac{p\left(y_{L,\sigma_i}|f(\sigma_{\le i}), y_{L,\sigma_{<i}}\right) p\left(f(\sigma_{\le i})|y_{L,\sigma_{<i}}\right)}{q(\sigma_i|\sigma_{<i},y_L)} \right] \right] \tag{21}$$

One problem with both Equation (18) and Equation (21) is that one needs to switch between two losses during training, i.e., 1) when sampling $t \sim \text{Uniform}[1 \ldots L]$, we optimize $\mathcal{L}_{\text{ELBO}}$, and 2) when $t = L + 1$, we optimize $\mathcal{L}^{\text{CB}}_{\text{Termination}}$, which may cause computational inefficiency. Moreover, Equation (21) implicitly assumes that EOS is always inserted at $L + 1$, i.e., $p(z_{L+1}|y_L) = \delta(L + 1)$. To address these two issues, we introduce an *augmented ELBO* below.

## C.1. Simplified Training Loss via Augmented ELBO

First, we observe that $\log p(x_{L+1} = \text{EOS}|y_L)$ can be viewed as a standard EOS prediction in FO-ARMs. In the context of modeling with Insertion Process, it simply requires the classifier to output EOS when it sees the complete sample $y_L$, and EOS is always predicted after $y_L = (x_1 \ldots x_L)$.

In fact, we can enforce the EOS token to be always sampled last through constraining the EOS logit outputted from the variational order policy network to a sufficiently small number $\epsilon$. For instance, if $y_{L+1} = (1, 2, 3, 4, \text{EOS})$, and assume that the variational policy logits corresponding to the first 4 tokens are $g = [10, 20, 30, 40]$, then the augmented logits become $\hat{g} = [10, 20, 30, \epsilon]$. In this way, with the Gumbel–Top-$k$ trick for sampling the variational order distribution $q_\theta$, the EOS dimension will always be sampled at last, e.g., at step $t = L + 1$, while the estimate of the ELBO is still unbiased when $t \in [1 \ldots L]$. In practice, we set $\epsilon = -1e7$ as constant during training.

Next, we show that we can include $\mathcal{L}^{\text{CB}}_{\text{Termination}}$ within $\mathcal{L}^{\text{CB}}_{\text{ELBO}}$ via this augmentation trick introduced above. To see this, we only consider the case of $t = L + 1$, in which only EOS is left to be generated, and all other tokens have been generated in the previous $L$ steps, the logits of which are set to $-\text{inf}$ correspondingly. For instance, in the example above $\hat{g} = [-\text{inf}, -\text{inf}, -\text{inf}, -\text{inf}, \epsilon]$. Therefore, we can see that $q(\sigma_{L+1} \mid \sigma_{<L+1}, y_L) = \text{Softmax}(\hat{g})_{L+1} = 1$, and $q(\sigma_{<L+1}|y_{L+1}) \approx 1$, which will be 1 numerically during training.

Actually $\mathcal{L}^{\text{CB}}_{\text{Termination}}$ can be augmented to $\mathcal{L}^{CB}_{\text{ELBO}}$ at $i = L + 1$ with the augmented $\hat{g}$ introduced above.

$$
\begin{aligned}
\mathcal{L}^{\text{CB},L+1}_{\text{ELBO}}| &= \mathbb{E}_{q(\sigma_{<L+1}|y_{L+1})} \left[ \mathbb{E}_{q(\sigma_{L+1}|\sigma_{<L+1}, y_{L+1})} \left[ \log \frac{p\left(y_{L+1,\sigma_{L+1}}|f(\sigma_{\leq L+1}), y_{L+1,\sigma_{<L+1}}\right) p\left(f(\sigma_{\leq L+1})|y_{L+1,\sigma_{<L+1}}\right)}{q(\sigma_{L+1}|\sigma_{<L+1}, y_{L+1})} \right] \right] \\
&= \mathbb{E}_{q(\sigma_{L+1}|\sigma_{<L+1}, y_{L+1})} \left[ \log p\left(y_{L,\sigma_{L+1}} = \text{EOS}\right) \right] - \text{KL}\left[ q(\sigma_{L+1}|\sigma_{<L+1}, y_{L+1}) \| p\left(f(\sigma_{\leq L+1})|y_{L+1,\sigma_{<L+1}}\right) \right] \\
&= \log p\left(y_{L+1,\sigma_{L+1}} = \text{EOS}\right) - \text{KL}\left[ q(\sigma_{L+1}|\sigma_{<L+1}, y_{L+1}) \| p\left(f(\sigma_{\leq L+1})|y_{L+1,\sigma_{<L+1}}\right) \right] \\
&= \mathcal{L}^{\text{CB}}_{\text{Termination}} - \text{KL}\left[ \delta(L+1) \| p\left(z_{L+1} = L + 1|y_{L+1,\sigma_{<L+1}}\right) \right]
\end{aligned}
\tag{22}
$$

In the last line we use the fact that, $z_{L+1} = f(\sigma_{\leq L+1}) = L + 1$, as with the augmentation trick, the EOS dimension will always be inserted last with the permutation orderings $\sigma$. As a result, the loss with the augmented ELBO becomes

$$
\begin{aligned}
\tilde{\mathcal{L}}^{\text{CB}} &= \sum_{i=1}^{L+1} \mathbb{E}_{q(\sigma_{<i}|y_L)} \left[ \mathbb{E}_{q(\sigma_i|\sigma_{<i}, y_L)} \left[ \log \frac{p\left(y_{L,\sigma_i}|f(\sigma_{\leq i}), y_{L,\sigma_{<i}}\right) p\left(f(\sigma_{\leq i})|y_{L,\sigma_{<i}}\right)}{q(\sigma_i|\sigma_{<i}, y_L)} \right] \right] \\
&= \mathcal{L}^{\text{CB}}_{\text{Termination}} + \mathcal{L}^{\text{CB}}_{\text{ELBO}} - \text{KL}\left[ \delta(L+1) \| p\left(z_{L+1} = L + 1|y_{L+1,\sigma_{<L+1}}\right) \right] \\
&= \mathcal{L}^{\text{CB}} - \text{KL}\left[ \delta(L+1) \| p\left(z_{L+1} = L + 1|y_{L+1,\sigma_{<L+1}}\right) \right]
\end{aligned}
\tag{23}
$$

For the KL term, the insertion policy is optimized against a Kronecker-delta point mass at $t = L + 1$, which is a low-cost optimization. In practice, it decays to less than $0.001$ within $20k$ training steps. As a result, $\tilde{\mathcal{L}}^{\text{CB}} \approx \mathcal{L}^{\text{CB}}$ as defined in Equation (21). Now we can see that with $\tilde{L}^{\text{CB}}$, no switch between $\tilde{L}^{\text{CB}}_{\text{Termination}}$ and $\tilde{L}^{\text{CB}}_{\text{ELBO}}$ is needed, unlike the case of policy-based termination, which requires an explicit EOS head.

Finally, similar to the case of the loss with policy-based termination, we implement the objective

$$
\frac{L + 1}{2} \left\{ \left( \log q(\sigma^1_{<i}|y_{L+1}) - \log q(\sigma^2_{<i}|y_{L+1}) \right) \text{stopgrad}[F^1 - F^2] + F^1 + F^2 \right\},
\tag{24}
$$

where for $j = 1, 2$

$$F^j = E_{q(\sigma_i^j|\sigma_{<i}^j, y_{L+1})} \left[ \log \frac{p\left(y_{L+1,\sigma_i^j}|f(\sigma_{\leq i}^j), y_{L+1,\sigma_{<i}^j}\right) p\left(f(\sigma_{\leq i}^j)|y_{L+1,\sigma_{<i}^j}\right)}{q(\sigma_i^j|\sigma_{<i}^j, y_{L+1})} \right]$$

## D. Experiment setup

### D.1. Maze Planning Dataset

Following Kim et al. (2025), we evaluate our model on a synthetic maze-planning dataset designed to test long-horizon, discrete-structure generation under partial conditioning. Each example is a grid-maze trajectory flattened into a 1D token sequence so that standard sequence models can be trained and used for inference on the same representation. For conditional generation, a subset of states along the trajectory are treated as ordered subgoals; for non-insertion baselines, we prepend these subgoal tokens to the input sequence.

Below we describe how mazes, paths, and train/validation/test splits are constructed.

**Maze graph construction.** We represent each maze as an unweighted grid graph over free cells. A maze of size $m$ is generated on a $(2m+1) \times (2m+1)$ grid using recursive division; cells with value 0 are traversable and cells with value 1 are walls. Each traversable cell becomes a node, with 4-neighbor (N,S,E,W) edges between adjacent free cells. Tokens are flattened cell indices $t = r \cdot W + c$ (vocabulary size $H \cdot W$), with separate special tokens for mask, bos, eos, and pad.

We generate paths in two ways. *Subgoals-first* samples $k$ distinct free cells as ordered subgoals; consecutive subgoals are connected by BFS shortest-path segments, which are concatenated (removing duplicate junction nodes) to form the final path.

*Path-first* samples start–goal pairs, computes a BFS shortest path, and then creates up to a fixed number of alternative simple paths by replacing a segment with a longer detour that avoids the original segment (and the rest of the path). This construction follows the setting used by FlexMDM, and we use it for the **imperfect** and **braided** maze variants. For each path, subgoals are selected by taking the endpoints plus $k-2$ uniformly sampled interior points along the path. Samples exceeding a maximum sequence length are discarded. Data splits are 80/10/10 (train/validation/test).

**Maze variants and parameters.** All mazes are produced by recursive division on a $(2m+1) \times (2m+1)$ grid; open cells form a 4-neighborhood graph.

- **Perfect mazes.** The base recursive-division generator produces tree-like mazes with no cycles.

- **Imperfect mazes.** Starting from a perfect maze, we identify wall cells that lie between two open cells (either N–S or E–W). These are candidate "doors" whose removal creates a local cycle. We open a random fraction $f$ of candidates, with $f \in [0, 1]$ and count $\lfloor f|\mathcal{C}| \rfloor$. This increases loopiness without fully destroying the global structure; we set $f = 0.3$.

- **Braided mazes.** Starting from a perfect maze, we identify dead ends (open cells with exactly one open neighbor). For a random fraction $b$ of these, we open a nearby wall to connect to an existing passage (two steps away), thereby eliminating the dead end and introducing a cycle. We clamp $b \in [0, 1]$ and open $\lfloor b|\mathcal{D}| \rfloor$ walls; we use $b = 1.0$ to remove as many dead ends as possible.

Unless stated otherwise, we use the default generator settings: maze size $m=20$ (grid $41 \times 41$), max path length 400.

### D.2. Star Graph Planning Dataset

We build the star-graph planning task adapted from Patel et al. (2025) but with different configuration. Each instance is a directed star-shaped graph with a single junction node shared across all arms. The model is given the graph structure as a randomly ordered list of directed edges (node pairs), followed by the source node, the goal node, and a special graph-BOS token. The model must then generate the directed path edges from source to goal. We condition on the entire prefix (edge list + source + goal + graph-BOS) and predict the path edge sequence.

**Dataset configuration used.** In all experiments we use the hard star-graph configuration: degree 5 with arm lengths sampled uniformly in $[6, 12]$ (nodes per arm), and a node vocabulary of size 56. The junction's position along each arm is randomized, yielding both incoming and outgoing edges with respect to the junction. The ground-truth path is the directed chain along one designated arm between its endpoints (source and goal).

**Evaluation metrics.** We report both exact-match and token-level correctness, as well as edit-based distances that quantify how far a predicted path deviates from the ground truth. Specifically, *sequence accuracy* (Seq. Acc.) measures the fraction of instances where the entire generated edge sequence exactly matches the gold path. *Token accuracy* (Tok. Acc.) computes the proportion of correctly predicted tokens under a position-wise comparison between the predicted and gold sequences (after aligning by position), reflecting partial correctness when only some edges are correct. To better capture discrepancies when the prediction has insertions/deletions or local mis-ordering, we additionally report the *Hamming distance* (Ham.) on the aligned sequences (lower is better), and the *Levenshtein edit distance* (Lev.), i.e., the minimum number of insertions, deletions, and substitutions required to transform the prediction into the gold sequence (lower is better). Together, these metrics characterize both exact planning success and the extent/type of failure modes.

### D.3. Training Hyperparameters for Planning Benchmark

We train on two planning datasets for 100 epochs with batch size 64 and no gradient accumulation. Optimization uses AdamW with learning rate $1 \times 10^{-4}$ and a cosine decay schedule with 1,000 warm-up steps. We use exponential moving average of parameters with decay 0.999 (updated every step, starting at step 200). The Transformer backbone has 12 layers, 8 attention heads, hidden size 128, dropout 0.1, and SwiGLU MLPs; we enable RoPE positional embeddings (Su et al., 2023) with base 10,000. The policy network uses 6 layers, 4 heads, hidden size 64, dropout 0.1, and RLOO loss with $M = 2$. Unless otherwise noted, all baselines use the same architecture. We implement MDM following AO-ARM, but replace the sampler with Euler integration using a number of sampling steps equal to the maximum sequence length.

### D.4. The GuacaMol Metrics

- (Preuer et al., 2018) introduced Fréchet ChemNet Distance (FCD) as a measure of how close distributions of generated samples are to the distribution of molecules in a reference set. The FCD is determined from the hidden representation of molecules in a neural network called ChemNet trained for capturing important chemical and biological features, similarly to the Fréchet Inception Distance (FID) (Heusel et al., 2017) in image generation. Note that, FCD is sample-size-dependent, and for all FCD evaluations against the GuacaMol benchmark, the standard in the literature is only using 10000 samples for both the generated and ground truth samples. Moreover, usually better generation performance yields smaller FCD, but the GuacaMol benchmark normalizes FCD, given by $S = \exp(-0.2 \cdot \text{FCD})$.

- KL divergence. For this task, a set of physicochemical descriptors calculated with the RDKIT for both the sampled and the reference set, and then the distributions of these descriptors is computed via kernel density estimation for continuous descriptors, or as a histogram for discrete descriptors. Finally, the KL divergence $D_{\text{KL},i}$ of each descriptor $i$ is aggregated through $S = \frac{1}{k} \sum_i^k \exp(-D_{\text{KL},i})$.

### D.5. Training Hyperparameters for GuacaMol

All baselines, except the graph-based methods DeFog (Qin et al., 2024) and Cometh (Siraudin et al., 2024), use a Transformer architecture. Our implementation is adapted from the `llama2.c` project (Touvron et al., 2023).[3] For FO-ARM and for the generative models $p_\theta$ in both LO-ARM and IP, we use Transformers based generative decoder with 18 attention layers. The variational inference encoder in LO-ARM and IP use a Transformer with 3 attention layers. Hyperparameters are reported in Table 5, and all experiments are run to convergence.

---

[3]https://github.com/karpathy/llama2.c

*Table 5.* Hyperparameter setup.

| Hyperparameter | ChEMBL/GuacaMol |
| --- | --- |
| Optimizer | AdamW |
| Scheduler | Cosine Annealing |
| Learning Rate for $p_\phi$ | $5 \cdot 10^{-4}$ |
| Learning Rate for $q_\theta$ | $5 \cdot 10^{-6}$ |
| Weight Decay | $1 \cdot 10^{-12}$ |
| EMA | 0.9999 |

# E. Visualization of Planning Trajectory

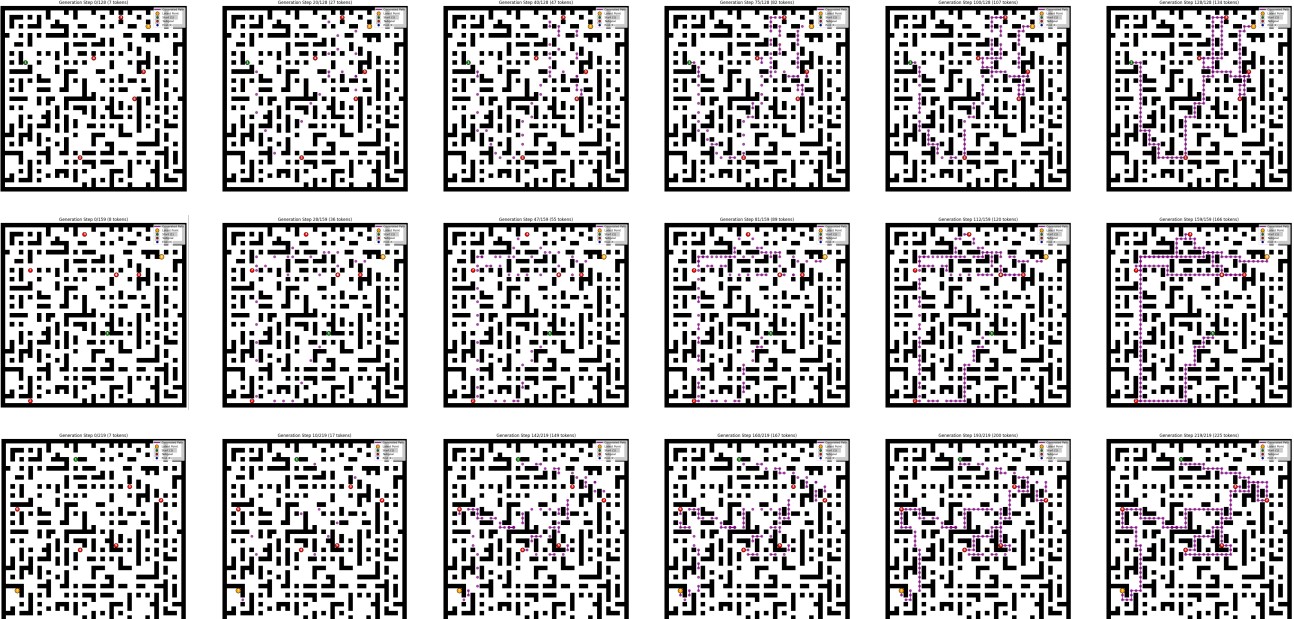

*Figure 6.* Visualization of Insertion Trajectory on Synthetic Maze Planning Dataset

# F. Gallery of Conditional Generations

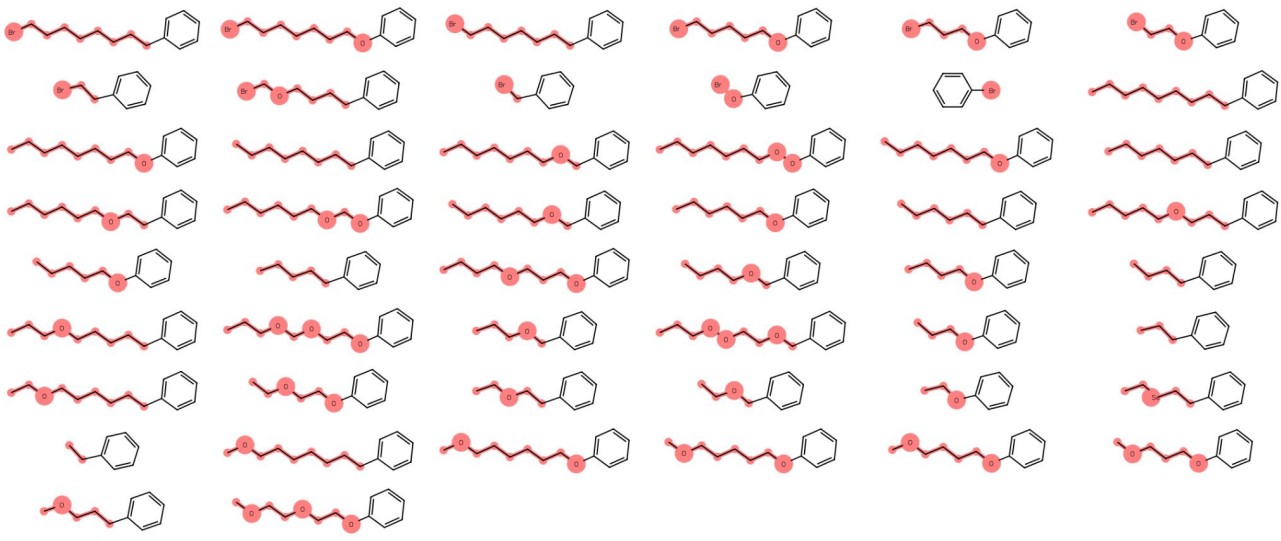

*Figure 7.* Generated molecules in fragment completion and decoration (Task 1). Generated atoms are highlighted in red.

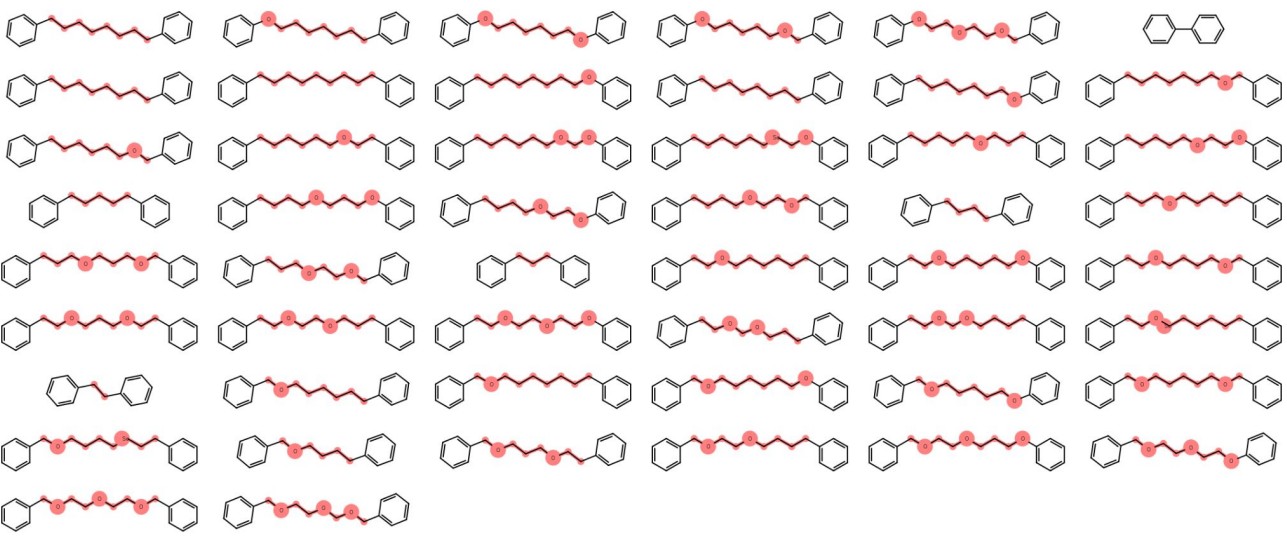

*Figure 8.* Generated molecules in linker design (Task 2). Generated atoms are highlighted in red.

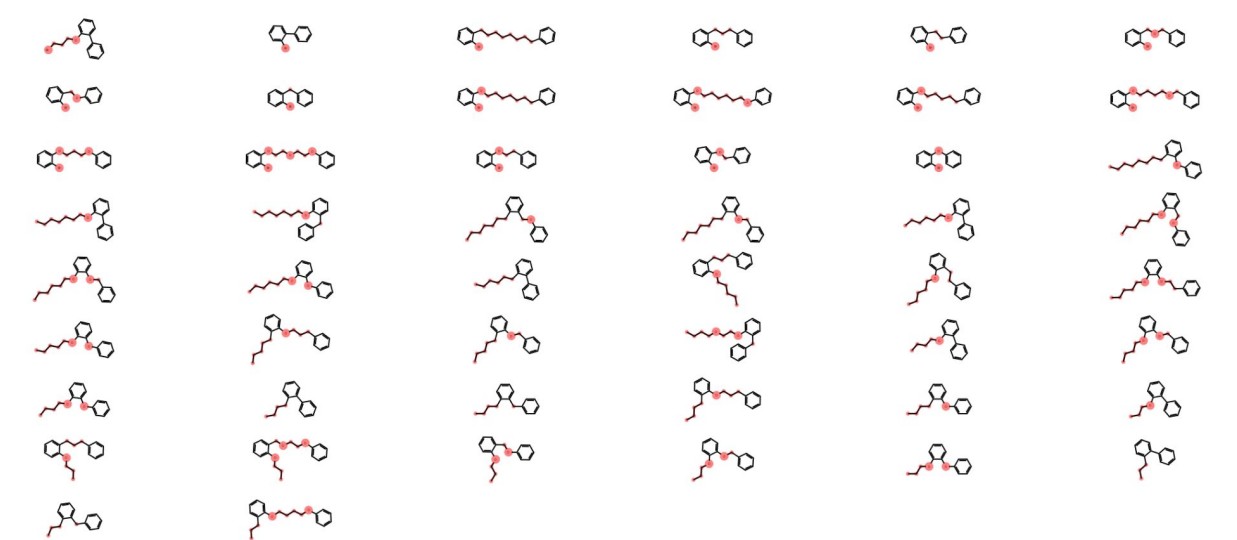

*Figure 9.* Generated molecules in linker design and partial fragment decoration (Task 3). Generated atoms are highlighted in red.

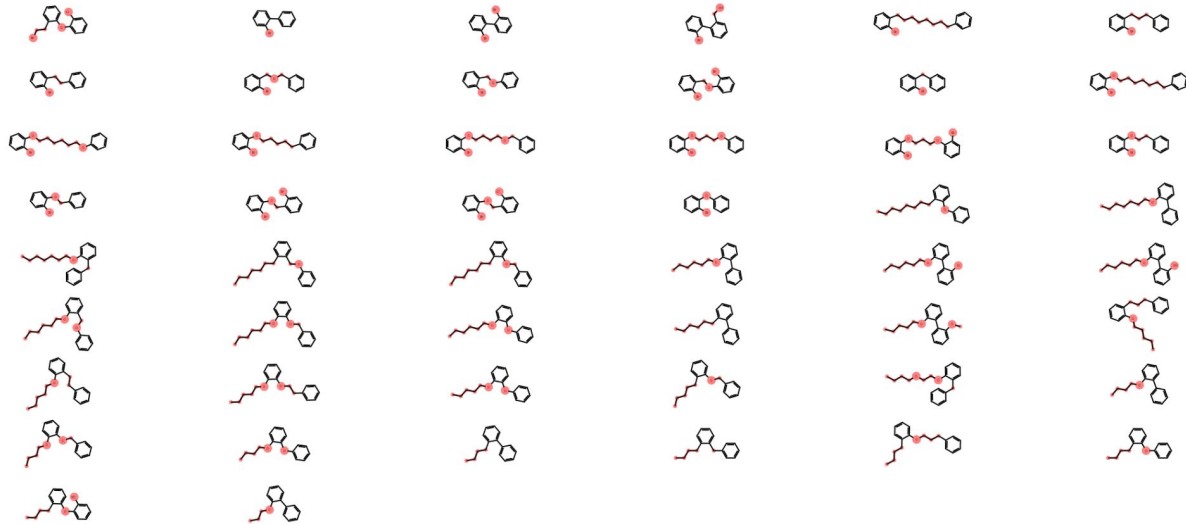

*Figure 10.* Generated molecules in linker design and full fragment decoration (Task 4). Generated atoms are highlighted in red.

