# OpenReview forum: "Variational Learning for Insertion-based Generation"
_ICML.cc/2026/Conference — ICML 2026 spotlight_

### Official Review · Reviewer_ScKq · 2026-03-09

**Soundness:** 3
**Presentation:** 2
**Significance:** 4
**Originality:** 3
**Overall Recommendation:** 5
**Confidence:** 3

**Summary:**

The paper introduces the Insertion Process (IP), a probabilistic framework for learning data-driven insertion orders in variable-length sequence generation. It tackles the limitations of standard monotonic left-to-right AR models and the rigid, fixed-canvas constraints typical of masked discrete diffusion models. The core theoretical contribution is a formalized bijection between relative insertion trajectories and global sequence permutations (Lemma 2.1). This reparameterization allows the authors to optimize the data likelihood via variational inference, treating the generation order as a latent variable. The latent order is modeled with a Plackett-Luce distribution, sampled via the Gumbel-Top-k trick, and optimized using a REINFORCE Leave-One-Out (RLOO) gradient estimator. Empirically, IP shows strong performance on constrained planning tasks (mazes, star graphs) and high novelty in molecular SMILES generation.

**Compliance With Llm Reviewing Policy:**

Affirmed.

**Final Justification:**

The paper presents a technically solid and original framework for learning insertion-based generation orders, with a particularly elegant theoretical formulation and strong significance for structured sequence generation. I weighed the strengths in soundness, novelty, and potential impact more heavily than the empirical and presentation weaknesses, especially since the experiments still demonstrate clear practical value. My main concerns were the clarity of the termination mechanism, the computational overhead of the training procedure, and the trade-off between novelty and distributional matching. The rebuttal addressed these concerns satisfactorily by clarifying the intended positioning of IP, explaining the fidelity–exploration trade-off, and committing to move important methodological details and limitations into the main paper. Given that I maintain my recommendation.

**Key Questions For Authors:**

1. Given the heavy training overhead of the Plackett-Luce encoder and RLOO optimization, how does IP compare to simply applying inference-time guidance on a standard discrete diffusion model to enforce these same structural constraints (e.g., in the maze or SMILES tasks)?

3. Can you provide more insight into the performance gap in distributional metrics (KL divergence and FCD) compared to FO-ARM on GuacaMol? Is this an inherent trade-off of the insertion process, or can the objective be tuned to prioritize empirical distribution matching over novelty?

**Limitations:**

The paper needs a dedicated technical limitations section. Specifically, the authors should explicitly discuss the computational overhead and variance introduced by the Plackett-Luce encoder and the RLOO gradient estimation during training. Furthermore, they need to directly address the observed trade-off between generating novel structures and matching empirical distributions, as evidenced by the lower FCD and KL divergence scores in the molecular generation tasks

**Strengths And Weaknesses:**

**Strengths**

&nbsp;

1. The mathematical formulation is highly elegant. The permutation-trajectory bijection (Lemma 2.1) neatly solves the shifting coordinate problem inherent to insertion-based generation. By mapping relative actions to a fixed global space, the authors make trajectory marginalization mathematically tractable for standard variational inference.

&nbsp;

2. Escaping the static canvas limitation common in discrete diffusion models represents a solid architectural step forward. The model's ability to actively learn generation priorities (e.g., building molecular rings before inserting atoms) without hand-crafted rules is practically very useful for structured domains like drug discovery.

&nbsp;


**Weaknesses**

&nbsp;

1. The empirical results show a distinct fidelity-exploration trade-off. While the IP generates highly novel molecular structures, it noticeably underperforms compared to a standard left-to-right FO-ARM on strict distributional matching metrics (KL divergence and FCD) in the GuacaMol benchmark.


&nbsp;

2.  The explanation of the termination mechanisms feels a bit disjointed. The classifier-based termination variant is heavily relegated to the appendix (Sec. C), even though it is the variant used for the main molecular generation experiments. Main section explaining the the method reads quite hard and can be rewriten

---

> ### Author Rebuttal · Authors · 2026-03-31
>
> We sincerely thank the reviewer for  recognizing the theoretical elegance of the permutation-trajectory bijection and its practical utility for structured domains. We address your questions below.
>
> **1. Inference-time guidance vs IP training**
>
> > "How does IP compare to simply applying inference-time guidance on a standard discrete diffusion model to enforce these same structural constraints?"
>
> We think IP and inference-time guidance are fundamentally different and complementary. IP is a generative model that learns the unconditional data distribution through an insertion-based process — constrained generation (e.g., generating from a partial sequence) emerges naturally from its formulation, without requiring any additional machinery. Inference-time guidance, on the other hand, is a post-hoc technique that requires training a separate task-specific classifier for each type of constraint, and applying strong guidance can shift the sampling distribution away from the learned data manifold.
>
> Importantly, these two approaches are not mutually exclusive — inference-time guidance can also be applied on top of IP to further steer generation. Due to this orthogonality, a direct benchmark comparison is not straightforward, but we consider exploring the combination of learned generation order with inference-time guidance as an interesting direction for future work.
>
> **2. Distributional metrics gap and tuning the objective**
>
> > "Can you provide more insight into the performance gap in distributional metrics (KL divergence and FCD) compared to FO-ARM on GuacaMol? Is this an inherent trade-off, or can the objective be tuned?"
>
> Thank you for raising this point. As discussed in our response to Reviewer nPYh, this gap reflects a fundamental expressiveness-tractability spectrum: AR models (fixed order, fixed length) → discrete diffusion (any order, fixed length) → IP (any order, flexible length). Each relaxation trades bound tightness for greater generative flexibility. The higher novelty and validity scores of IP demonstrate the value of this flexibility.
>
> Regarding tuning: yes, the objective can be adjusted. For instance, decreasing the sampling temperature during the Gumbel-Top-k process encourages the model to concentrate on fewer, higher-probability trajectories, which would potentially improve distributional matching at the cost of reduced novelty. We will discuss this in the revised manuscript.
>
> **3. Presentation of termination mechanisms and technical limitations**
>
> > "The classifier-based termination variant is heavily relegated to the appendix... The paper needs a dedicated technical limitations section."
>
> We completely agree. In the revised manuscript, we will:
> 1. **Revise and move the section about classifier-based termination into the main text**, as it is the core mechanism enabling our SMILES results.
> 2. **Add a dedicated Technical Limitations section** discussing the training overhead of the PL encoder and RLOO estimation, and a discussion of the fidelity-exploration tradeoff.
>
> Thank you again for these constructive suggestions — they will significantly improve the paper's presentation.

---

> > ### Author Rebuttal · Reviewer_ScKq · 2026-04-03
> >
> > Thank you for providing further clarification.

---

### Official Review · Reviewer_eq3k · 2026-03-13

**Soundness:** 3
**Presentation:** 3
**Significance:** 3
**Originality:** 3
**Overall Recommendation:** 5
**Confidence:** 3

**Summary:**

The paper tackles the limitations of existing non-monotonic sequence generation models, such as static canvas and order agnosticism, which are critical issues in various domains like planning and molecular generation.

**Compliance With Llm Reviewing Policy:**

Affirmed.

**Final Justification:**

The author has clearly anwered my questions

**Key Questions For Authors:**

Question 1: The computational complexity of the proposed method, particularly due to the permutation-based variational inference, might be a concern for practical applications. Have you explored any approximations or efficient implementations to mitigate this issue?


Question 2: The paper focuses on policy-based termination. Have you considered or explored classifier-based termination in more detail, and if so, what were the key differences in performance or behavior between the two approaches?

Question 3: The paper lacks ablation studies to assess the impact of individual components of the model. Have you conducted any preliminary experiments to evaluate the contribution of the Plackett-Luce model or the RLOO gradient estimator?

Question 4: The paper mentions the potential applicability of the proposed approach to protein design. Have you conducted any exploratory experiments or analysis in this direction?

**Limitations:**

Yes

**Strengths And Weaknesses:**

Strengths: The paper introduces a novel probabilistic framework for learning insertion order in variable-length insertion models, offering a fresh perspective on sequence generation.
Weaknesses:
1)Because the proposed method requires variational inference based on permutations, its computational complexity may be higher than that of simpler models, especially when dealing with long sequences. Furthermore, the issue of computational efficiency is not discussed in detail.
2)The paper compares the proposed method primarily with other non-monotonic generation models and does not include comparisons with state-of-the-art methods for monotonic sequence generation. A more comprehensive comparison would strengthen the evaluation of the proposed approach.
3)The paper focuses on policy-based termination and provides limited exploration of classifier-based termination.
4)The paper lacks ablation studies to assess the impact of individual components of the model, such as the Plackett-Luce model or the RLOO gradient estimator.

---

> ### Author Rebuttal · Authors · 2026-03-31
>
> We sincerely thank the reviewer for the thorough and constructive feedback. Your suggestions have helped us strengthen the paper substantially. We address your questions below.
>
> **1. Computational complexity**
>
> > "The computational complexity of the proposed method, particularly due to the permutation-based variational inference, might be a concern. Have you explored any approximations or efficient implementations?"
>
> We would like to clarify that the permutation-based variational inference (the PL encoder q_θ) is only used during training and adds zero inference cost. At inference time, the per-step FLOPs of IP (952.4M) are within 1.72% of a standard autoregressive model (936.2M) with the same backbone, with the slot prediction module itself contributing less than 0.004% overhead. We will add a dedicated Technical Limitations section to discuss this clearly.
>
> **2. Monotonic sequence generation baselines**
>
> > "The paper compares primarily with other non-monotonic generation models and does not include comparisons with state-of-the-art methods for monotonic sequence generation."
>
> We would like to clarify that FO-ARM (Fixed-Order Autoregressive Model), which serves as one of our primary baselines throughout all experiments, is a standard monotonic left-to-right generation model. IP consistently outperforms FO-ARM on planning tasks (e.g., 97.9% vs 27.2% on Perfect Hard mazes), demonstrating clear advantages over monotonic generation when structural ordering matters. We will emphasize this distinction more clearly in the revised experimental setup.
>
> **3. Classifier-based termination**
>
> > "Have you considered or explored classifier-based termination in more detail?"
>
> Yes, classifier-based termination is in fact the variant used for our main molecular generation experiments (SMILES). We provide a direct comparison on GuacaMol below:
>
> | Termination | Validity | Uniqueness | Novelty | FCD |
> |-------------|----------|-----------|---------|-----|
> | Policy-based | 97.2 | 99.8 | 97.8 | 88.2 |
> | Classifier-based | 97.4 | 99.9 | 98.3 | 89.2 |
>
> Classifier-based termination consistently outperforms across all metrics. We attribute this to the decoupling of the termination decision from the insertion decision — in policy-based termination, the EOS signal competes with insertion slots in the same softmax, which can lead to premature termination, whereas classifier-based termination evaluates completeness independently. We agree with the reviewer that relegating this to the appendix was a presentation issue and will move the discussion into the main text in the revised manuscript.
>
> **4. Ablation studies (Plackett-Luce and RLOO)**
>
> > "Have you conducted any preliminary experiments to evaluate the contribution of the Plackett-Luce model or the RLOO gradient estimator?"
>
> Thank you for raising this point, we have run these ablations.
>
> **Plackett-Luce vs. Uniform (order-agnostic):** As shown in our response to Reviewer M8GY, switching from learned PL ordering to uniform random ordering causes a dramatic drop on structurally complex tasks (e.g., 97.9% → 4.8% on Perfect Hard mazes), while simpler tasks show smaller gaps. This confirms that learned ordering is critical when structural dependencies are complex.
>
> **M sweep (M=2,3,4):** We ablated the RLOO sample size M on the molecule generation (GuacaMol) benchmark under a fixed computational budget (M × batch_size = constant). With M=2/BS=1024 and M=4/BS=512 (same budget), performance is nearly identical. Reducing the budget (M=3/BS=512) leads to a slight drop as expected. This indicates that increasing M provides no additional benefit when the total compute is held constant — simply increasing batch size with M=2 is the most effective strategy. See our response to Reviewer nPYh for the full ablation table and figure.
>
> **5. Protein design applicability**
>
> > "Have you conducted any exploratory experiments or analysis in protein design?"
>
> Thank you for your suggestions! While we believe variable-length insertion is theoretically well-suited for protein design (e.g., modeling variable-length loops in a fixed scaffold), rigorous empirical validation in this domain requires scaling to large pre-trained structural models, which exceeds our current computational resources. We leave this as a promising future application and are actively exploring it.
>
> We hope to have addressed the reviewer questions and concerns: if so, we'd be grateful if the reviewer might consider increasing their score.

---

### Official Review · Reviewer_M8GY · 2026-03-13

**Soundness:** 3
**Presentation:** 4
**Significance:** 3
**Originality:** 3
**Overall Recommendation:** 5
**Confidence:** 5

**Summary:**

The paper propose a simple variational approach for learning an insertion based sequence generation policy consisting of both the generation order and token selection. Since the generation order is not observable in the data, the paper proposes to minimize a variational lower bound that uses a ground truth sequence conditional variational permutation distribution (which can be mapped to an insertion order). This is similar to [1], but instantiated in the context of insertion based generation. Theoretical justification of the learning procedure is provided using the ELBO. Preliminary empirical investigation is conducted on two synthetic planning tasks (path finding on star graphs and maze), both of which can only be solved well if the generative method can generate variable length sequences as well as discover favorable generation orders. Finally, the usefulness of the approach is demonstrated on unconditional as well as constrained molecule generation using SMILES strings.

[1] Wang, Z., Shi, J., Heess, N., Gretton, A., and Titsias, M. K. Learning-order autoregressive models with application to molecular graph generation. In Forty-second International Conference on Machine Learning, 2025b.

**Compliance With Llm Reviewing Policy:**

Affirmed.

**Final Justification:**

The authors have answered my questions clearly and in the process have improved the empirical analysis in the paper. I'm updating my recommendation to accept.

**Key Questions For Authors:**

1. I think there could be two additional baselines that would really shine light on the improvements offered by the proposed approach.

(a) One order agnostic insertion process baseline as discussed in the related work. The closest ones would be [1] or [2], or perhaps even more comparable would to simply switch off $q_\theta$ in Algorithm 1, and sample uniformly random permutations. To my understanding, this would be the same as [2] but with slightly different termination criteria.

(b) Some existing order learning methods that also use insertion like process, for instance [3], would also provide a good comparison.

I think both these baselines are important to establish the usefulness of the approach. Is there any specific reason for not having these baselines?

2. Order agnostic training can make the model robust for constrained generation, for example, when a given partial sequence has low or negligible probability under $q_\theta$, the generator $p_\phi$ is not used to seeing such a partial sequence (might call it exposure bias [5] ). It will be informative to construct such test set of low probability partial sequences to stress test the generator and check its robustness. This is one more reason to have a order agnostic insertion baseline.

3. Having an additional model $q_\theta$ and additional forward passes can show down the training. However, not training on all order can make the procedure statistically efficient. It will be informative to compare the training efficiency of the proposed approach with an order agnostic insertion model.

**Minor clarification questions**:
1. From figure 1 and the explanation in section 2.4, it clear how $h_i$ is obtained, but how is $w_z$ used in Eq. 15 obtained?

2. Do the networks $q_\theta$ and $p_\phi$ share parameters (eg. a shared transformer backbone)?



# References

[1] Stern, M., Chan, W., Kiros, J., and Uszkoreit, J. Insertion transformer: Flexible sequence generation via insertion operations. In Chaudhuri, K. and Salakhutdinov, R. (eds.), Proceedings of the 36th International Conference on Machine Learning, volume 97 of Proceedings of Machine Learning Research, pp. 5976–5985. PMLR, 09–15 Jun 2019.

[2] Patel, D., Sahoo, A., Amballa, A., Naseem, T., Rudner, T. G. J., and McCallum, A. Insertion language models: Sequence generation with arbitrary-position insertions, 2025.

[3] Gu, J., Liu, Q., and Cho, K. Insertion-based decoding with automatically inferred generation order. Transactions of the Association for Computational Linguistics, 7:661676, 2019a.

[4] Li, X., Trabucco, B., Park, D. H., Luo, M., Shen, S., Darrell, T., and Gao, Y. Discovering non-monotonic autoregressive orderings with variational inference, 2021.

[5] Schmidt, F. (2019, November). Generalization in generation: A closer look at exposure bias. In Proceedings of the 3rd Workshop on Neural Generation and Translation (pp. 157-167).

**Limitations:**

yes

**Strengths And Weaknesses:**

**Soundness**: The proposed approach is theoretically sound and supported well. On the empirical side, I think two additional baselines are required (please see the questions section below).

**Presentation**: All the sections of the paper are very well written and easy to follow. The discussion on the related works is quite comprehensive and situates the proposed method appropriately within the context of existing literature. The experiments are clearly described and should be reproducible.

**Originality and Significance**: The is highly influenced by [1] and shares many similarities. For example, both define variational generation order over global permutations and use RLOO style gradient estimation. However, the key difference is that the proposed approach uses insertion policies (with a termination criteria) as opposed to LO-ARM policy.


[1] Wang, Z., Shi, J., Heess, N., Gretton, A., and Titsias, M. K. Learning-order autoregressive models with application to molecular graph generation. In Forty-second International Conference on Machine Learning, 2025b.

---

> ### Author Rebuttal · Authors · 2026-03-31
>
> We sincerely thank the reviewer for evaluating our paper as theoretically sound and well-written. We address your suggestions below.
>
> **1. Order-agnostic and existing order-learning baselines**
>
> > "(a) One order agnostic insertion process baseline... (b) Some existing order learning methods, for instance Gu et al. 2019a."
>
>  **(a)** We ran exactly the ablation described — switching off the learned permutation and sampling uniformly random insertion orders (Random Order IP). Results on the planning benchmarks:
>
> | Method | Braided (Easy) | Braided (Med) | Braided (Hard) | Perfect (Easy) | Perfect (Med) | Perfect (Hard) |
> |--------|-------|------|------|-------|------|------|
> | Random Order IP | 97.3 | 97.8 | 99.8 | 20.0 | 11.1 | 4.8 |
> | IP (Ours) | 100.0 | 100.0 | 99.9 | 99.4 | 98.1 | 97.9 |
>
> On Braided mazes, Random Order IP already performs well, suggesting the insertion framework itself provides value. However, on Perfect mazes — where structural dependencies are much more complex — the gap is striking (97.9% vs 4.8% on Hard), demonstrating that learned insertion order is critical.
>
> **(b)** We implemented the order-learning method of Gu et al. (2019a) on our planning benchmarks. The results were actually near-zero across all difficulties. We attribute this to their non-parametric approach, which relies on beam search to find optimal insertion orders. The search space grows combinatorially with sequence length, making it unable to find stable orders on our long-sequence maze planning tasks and leading to highly unstable training. This highlights the advantage of our parametric policy approach, which amortizes order prediction and scales gracefully to longer sequences.
>
> **2. Exposure bias and robustness**
>
> > "It will be informative to construct such test set of low probability partial sequences to stress test the generator."
>
> Thank you for raising this excellent point! When conditioned on partial sequences that fall outside the distribution of the learned policy q, the generator may indeed produce degraded outputs — this is why our current constrained generation results (Figures 8, 9) are presented as demonstrations, as the primary focus of our paper is unconditional generation. That said, we have a principled solution: incorporating constraints directly during training by conditioning the policy as q(σ|x, c), where c is a condition mask that remains fixed throughout generation. In fact, our planning experiments already adopt this approach — the maze start, end, and subgoal conditions are provided as training-time constraints, which is why IP achieves near-perfect performance on these tasks without exposure bias issues. Extending this to general conditional generation (e.g., molecular scaffold completion) is a promising direction for future work, which we are actively exploring.
>
> **3. Training efficiency comparison**
>
> > "It will be informative to compare the training efficiency of the proposed approach with an order agnostic insertion model."
>
> We compared the validation loss curves of IP (learned order) and Random Order IP on the maze planning benchmark during training. As shown in [Figure: Training Efficiency](https://anonymous.4open.science/r/icml-rebuttal-figures-276D/training_efficiency.png), IP consistently achieves lower validation loss at every checkpoint, converging to 6.37 compared to 7.42 for Random Order IP — a ~14% gap. This suggests that learning the insertion order not only improves generation quality but also makes training more statistically efficient, as the model concentrates capacity on high-probability trajectories rather than spreading it across all possible orderings.
>
> **4. Minor clarifications (w_z in Eq. 15 and parameter sharing)**
>
> > "How is w_z used in Eq. 15 obtained? Do the networks q_θ and p_φ share parameters?"
>
> w_z in Eq. 15 is a learnable weight vector in the location head of the generative decoder. It projects the contextualized slot embeddings h_{k-1} (produced by the Transformer decoder) into scalar logits, which are then normalized via softmax to produce the insertion slot distribution p_φ(z_i = k | y_{i-1}). It is trained end-to-end with the rest of the decoder parameters.
>
> Regarding parameter sharing: q_θ (the Plackett-Luce inference encoder) and p_φ (the generative decoder) do not share parameters. The encoder is a bidirectional Transformer that processes the full target sequence to produce ordering scores, while the decoder is a bidirectional Transformer that operates on partial sequences. Despite both being bidirectional, the encoder processes the full target sequence to score global ordering, whereas the decoder sees only the current partial sequence to predict the next insertion — their inputs and objectives are fundamentally different, making parameter sharing impractical. We will clarify this in the revised manuscript.
>
> We hope to have addressed the reviewer questions and concerns: if so, we'd be grateful if the reviewer might consider increasing their score.

---

> > ### Author Rebuttal · Reviewer_M8GY · 2026-04-02
> >
> > Thank you for answering my questions. I still have the following suggestions:
> >
> > 1. Have the random order IP baseline for Table 3 and Table 4 as well.
> > 2. If possible, compare the training progress for random IP vs IP using eval metrics vs wall clock time. This is more informative than val loss vs training steps plot because
> > (1) The ELBO (and hence val loss) will generally be tighter for IP compared to random order. Therefore checking evaluation metric will be a better indicator of the real training progress.
> > (2) Since IP uses two networks, each training step is more expensive compared to random IP, which is totally fine because training efficiency is not as important if the final performance is better, but having such a graph will provide a clearer picture about the proposed method to the readers.
> >
> > I'd be happy to raise my score if these two concerns are addressed.

---

> > > ### Author Response · Authors · 2026-04-06
> > >
> > > We thank the reviewer for their thoughtful follow-up and for indicating willingness to raise the score. We address both concerns below, referring to Any-Order IP as the random-order baseline and IP as our learned-order model.
> > >
> > > > 1. Have the random order IP baseline for Table 3 and Table 4 as well.
> > >
> > > **Table 3 (updated): Unconditional molecule generation on GuacaMol.**
> > >
> > > To provide a clearer picture of the effect of learned insertion order, we report compound metrics (Validity & Uniqueness, Validity & Uniqueness & Novelty) alongside individual ones:
> > >
> > > | Method | Validity | V & U | V & U & N | FCD |
> > > | :--- | :---: | :---: | :---: | :---: |
> > > | Any-Order IP | 88.2 | 88.2 | 87.7 | 79.1 |
> > > | IP (policy-based termination) | 97.2 | 97.0 | 94.9 | 88.2 |
> > > | IP (classifier-based termination) | **97.4** | **97.3** | **95.6** | **89.2** |
> > >
> > > Learning the insertion order yields consistent and substantial improvements across all metrics: +9.2% in validity, +9.1% in V&U, +7.9% in V&U&N, and +10.1 in FCD over the Any-Order baseline. This demonstrates that the learned order is not merely improving one metric at the expense of others, but lifting overall generation quality.
> > >
> > > **Table 4 (updated): Conditional generation tasks.**
> > >
> > > | Task | Validity | | Valid & Unique | |
> > > | :--- | :---: | :---: | :---: | :---: |
> > > | | IP | Any-Order IP | IP | Any-Order IP |
> > > | Fragment completion and decoration | **99.8** | 40.9 | **10.4** | 5.1 |
> > > | Linker design | **99.5** | 50.6 | **13.1** | 8.2 |
> > > | Linker design w/ partial frag. decoration | **99.9** | 42.8 | **24.0** | 20.6 |
> > > | Linker design w/ full frag. decoration | **99.9** | 41.3 | **34.2** | 31.1 |
> > >
> > > Each model generated 1024 molecules per task. Learning-Order IP achieves higher validity across all tasks and produces more valid and unique molecules per generation budget.
> > >
> > > > 2. The training progress for random IP vs IP using eval metrics vs wall clock time.
> > >
> > > We include FCD, validity, and novelty vs. wall-clock time comparisons below:
> > >
> > > [Figure: FCD vs wall-clock time](https://anonymous.4open.science/r/icml-rebuttal-figures-276D/fcd_comparison.png)
> > >
> > > [Figure: Validity vs wall-clock time](https://anonymous.4open.science/r/icml-rebuttal-figures-276D/validity_comparison.png)
> > >
> > > [Figure: Novelty vs wall-clock time](https://anonymous.4open.science/r/icml-rebuttal-figures-276D/novelty_comparison.png)
> > >
> > > In terms of FCD and validity, Learning-Order IP consistently converges faster in wall-clock time despite having higher per-step cost due to the additional policy network. This demonstrates that the learned order not only improves final performance but also accelerates training convergence — the model reaches the same FCD and validity levels as Any-Order IP in less wall-clock time. For novelty, both models achieve >98% (well above all other baselines in Table 3). Note that the higher novelty of Any-Order IP is partly attributable to its lower validity — invalid SMILES strings are inherently dissimilar to training molecules, inflating the novelty metric. When considering only valid and novel molecules jointly (V&U&N in Table 3), Learning-Order IP achieves 95.6% vs 87.7%, indicating superior meaningful novelty.
> > >
> > > We believe these results fully address both concerns. We remain available for any further discussion and thank the reviewer again for the constructive engagement.

---

### Official Review · Reviewer_nPYh · 2026-03-13

**Soundness:** 3
**Presentation:** 3
**Significance:** 4
**Originality:** 3
**Overall Recommendation:** 4
**Confidence:** 3

**Summary:**

This paper proposes the Insertion Process (IP), a probabilistic framework for variable-length sequence generation via learned insertion orders. The key technical insight is a bijection between insertion trajectories and permutations of target indices (Lemma 2.1), which converts the intractable marginalization over trajectories into a sum over permutations. This enables variational training with a Plackett-Luce posterior over orders, optimized via Gumbel-Top-k sampling and RLOO. Experiments on maze/star-graph planning and SMILES molecule generation show strong improvements over both fixed-order and order-agnostic baselines.

**Compliance With Llm Reviewing Policy:**

Affirmed.

**Key Questions For Authors:**

1. Distributional metrics on GuacaMol. FO-ARM still wins on KL div and FCD. The paper frames this as a "flexibility-fidelity tradeoff" which is a reasonable interpretation, but one could also read this as IP having a looser fit to the data distribution. It would help to report test NLL or ELBO values directly so we can assess whether this gap is due to the variational bound being loose or the model genuinely exploring more.

2. For conditional generation experiments (figure 4), it's hard to interpret without baselines. I feel other non-monotonic baselines can presumably handle those tasks too.

3. For RLOO, paper uses M=2. Does increasing M help with larger sequences?

4. Minor: What is the wall-clock generation time compared to MDM and FO-ARM on the planning benchmarks?

5. For the SMILES experiments, what happens if you initialize the insertion order to left-to-right (i.e., warm-start q_theta) vs random initialization? Does the learned order always converge to the skeleton-first pattern?

**Limitations:**

1. Generation under IP is inherently sequential. Each token requires a full decoder forward pass with no opportunity for parallel unmasking. The paper doesn't discuss this computational tradeoff, and for longer sequences the gap in inference cost could be substantial. Relatedly, the PL variational posterior assigns static scores from a single encoder pass, meaning the ordering decision for position j doesn't condition on which positions were already selected. For sequences where ordering decisions are highly coupled, this factorization could yield a loose bound.

2. Minor: Section 2.4: "illurated" to "illustrated"

**Strengths And Weaknesses:**

1. The permutation reparameterization is mathematically clean. Converting trajectory-dependent relative slot indices to a fixed latent space over permutations is a nice observation, and the fact that f(sigma_<=i) depends only on the prefix is well-exploited for variance reduction.

2. The empirical gains on planning are quite convincing, especially on Perfect mazes.

3. The SMILES generation analysis is insightful. The fact that IP discovers a consistent "skeleton-first, atoms-second" strategy without any domain-specific inductive bias is a compelling demonstration that the variational order learning is doing something meaningful.

---

> ### Author Rebuttal · Authors · 2026-03-31
>
> We sincerely thank the reviewer for the constructive feedback and for recognizing the mathematical cleanliness of our permutation reparameterization. We address specific questions and concerns below.
>
> **1. Distributional metrics on GuacaMol and the flexibility-fidelity tradeoff**
>
> We have computed the test NLL/ELBO for both models: FO-ARM achieves a test NLL of 31.9, while IP achieves an ELBO of ≤ 37.8. This gap reflects an expressiveness-tractability spectrum in discrete generative models. Autoregressive models adopt the most constrained generation mode — fixed left-to-right order with fixed length — yielding tight, tractable likelihoods. Discrete diffusion models (e.g., MDLM, MD4 [Shi et al., 2024]) relax the ordering constraint by allowing any-order generation, but at the cost of a looser ELBO — as noted by Shi et al., masked diffusion models are not yet competitive with AR models on likelihood precisely because AR models can better leverage model capacity by committing to a single order. Our Insertion Process further relaxes the generation process by additionally allowing flexible-length sequences, introducing yet another degree of freedom. Each step along this relaxation spectrum trades bound tightness for greater generative flexibility.
> Critically, the generation quality metrics demonstrate the value of this flexibility: IP achieves substantially higher novelty and validity, and dominates on planning tasks where structural ordering matters. We will incorporate this analysis with detailed NLL/ELBO comparisons in the revised manuscript.
>
> **2. Clarification on conditional generation experiments**
>
>  Figure 4 demonstrates the learned unconditional insertion order; we believe the reviewer may be referring to conditional generation in Figures 8–9 (quantitative results in Table 4).
>
> These conditional insertion tasks are precisely where IP's flexible-length capability becomes essential. Fixed-length baselines such as MDM cannot directly perform conditional insertion where the inserted segment length is unknown. Inference-time guidance is a possible workaround, but requires training a separate classifier for each type of conditioning constraint, making it impractical as a general-purpose baseline for benchmarking. In contrast, IP handles variable-length conditional insertion natively through its learned termination mechanism, without any task-specific modifications. We will clarify this distinction in the revised manuscript.
>
> **3. RLOO M sweep**
>
> We conducted an ablation studying the effect of RLOO sample size M under a fixed computational budget (M × batch_size = constant, since each RLOO update requires M forward passes through both p and q networks):
>
> | Setup | M | Batch Size | Budget (M × BS) |
> |---|---|---|----|
> | 1 | 2 | 1024 | 2048 |
> | 2 | 4 | 512 | 2048 |
> | 3 | 3 | 512 | 1536 |
>
> Setup 3 intentionally uses a smaller total budget to introduce additional sampling variance. As shown in [Figure: M Ablation](https://anonymous.4open.science/r/icml-rebuttal-figures-276D/M_ablation.png), Setups 1 and 2 perform nearly identically, while Setup 3 lags slightly behind as expected due to its reduced budget. This indicates that under a fixed computational budget, increasing M does not provide additional benefit — the variance reduction from more samples is offset by the smaller batch size. In practice, simply increasing the batch size with M=2 is the most effective strategy. We will include this ablation in the appendix.
>
> **4. Inference wall-clock time**
>
> Since different methods generate sequences of different lengths, we report average per-step FLOPs for a fair comparison. With L=125, d=128, N=12, the per-step inference FLOPs of IP (952.4M) are within **1.72%** of an autoregressive model with the same backbone (936.2M). The slot prediction module itself contributes less than 0.004% overhead (32K FLOPs vs 952M total). Furthermore, the policy network that learns the generation order is only used during training and adds zero inference cost.
>
> **5. SMILES initialization (L2R warm-start vs random)**
>
> We thank the reviewer for this interesting suggestion! We initialized the PL encoder with a left-to-right bias (adding a linearly decreasing bias to the scoring logits). Our preliminary results show that the learned order largely remains left-to-right, with occasional non-monotonic deviations (see [Figure: L2R Init](https://anonymous.4open.science/r/icml-rebuttal-figures-276D/l2r_init.png)). This suggests that L2R initialization creates a strong local minimum, and the skeleton-first pattern from random initialization is not the unique optimum but reflects initialization bias. The loss landscape for insertion order learning appears to have multiple local optima. We will include this analysis in the revised manuscript.
>
> **6. Formatting typo**
> Thank you for catching this! We will correct the typo in the revised manuscript.
>
> We'd be grateful if the reviewer might consider increasing their score if these updates are satisfactory.

---

> > ### Author Rebuttal · Reviewer_nPYh · 2026-04-03
> >
> > Thanks for the rebuttal. My questions are resolved.

---

### Decision · Program_Chairs · 2026-04-30

**Decision:**

Accept (spotlight)

**Comment:**

This submission introduces the Insertion Process, a probabilistic framework for variable-length sequence generation that learns where to insert tokens, what to insert,c and when to terminate. Relative to previous work on insertion-based generation, the main complexity introduced by variable length sequences is that a model cannot rely on a fixed set of indices to decide where to insert. The workaround that the authors identify (Lemma 2.1) is that during training, insertion order can be represented using a randomly sampled permutation, resulting in a trajectory that is amenable to ELBO-based training with Packett-Luce based order policy (via REINFORCE-LOO). Dealing with variable cardinality in generative models is harder than it looks, and this paper formulates an elegant approach in the context of sequence generation. Reviewers were happy with author responses and are in consensus. This is a clear accept.